# Reduction of precious metal ions in aqueous solutions by contact-electro-catalysis

Yusen Su [1,2,10], Andy Berbille [1,2,10], Xiao-Fen Li[1,3,10], Jinyang Zhang[1,2,10], MohammadJavad PourhosseiniAsl [1,4], Huifan Li[1,5], Zhanqi Liu[1,6], Shunning Li [7], Jianbo Liu [3], Laipan Zhu [1,2] ✉ & Zhong Lin Wang [1,2,8,9] ✉

Precious metals are core assets for the development of modern technologies in various fields. Their scarcity poses the question of their cost, life cycle and reuse. Recently, an emerging catalysis employing contact-electrification (CE) at water-solid interfaces to drive redox reaction, called contact-electro-catalysis (CEC), has been used to develop metal free mechano-catalytic methods to efficiently degrade refractory organic compounds, produce hydrogen peroxide, or leach metals from spent Li-Ion batteries. Here, we show ultrasonic CEC can successfully drive the reduction of Ag(ac), $Rh^{3+}$, $[PtCl_4]^{2-}$, $Ag^+$, $Hg^{2+}$, $Pd^{2+}$, $[AuCl_4]^-$, and $Ir^{3+}$, in both anaerobic and aerobic conditions. The effect of oxygen on the reaction is studied by electron paramagnetic resonance (EPR) spectroscopy and ab-initio simulation. Combining measurements of charge transfers during water-solid CE, EPR spectroscopy and gold extraction experiments help show the link between CE and CEC. What's more, this method based on water-solid CE is capable of extracting gold from synthetic solutions with concentrations ranging from as low as 0.196 ppm up to 196 ppm, reaching in 3 h extraction capacities ranging from 0.756 to 722.5 mg g$^{-1}$ in 3 h. Finally, we showed CEC is employed to design a metal-free, selective, and recyclable catalytic gold extraction methods from e-waste aqueous leachates.

Precious metals (PMs) are key components of various technologies, including electronics, catalysis, energy storage, medical implant, etc[1]. The recent awareness around the importance of considering the life cycles and reusability of PMs stems from their inherent scarcity, and their high demand in modern economies[2]. To address these challenges, industrial and public research sectors have adopted various strategies ranging from replacing PMs with earth crust abundant materials[3] to developing physical and chemical[4] recycling methods. In the industry, the most prevalent methods for the treatment of e-waste are based on pyrometallurgical and hydrometallurgical processes[5]. These strategies are simple and cost-effective on a large scale but come with their own sets of drawbacks. Alternative, more environmentally friendly, and selective methods to recover metals, especially gold, from e-waste water are highly sought after and intensively investigated. To this regard, public research in this area mostly focuses on methods based on solvent extraction[6], selective precipitation[7], or

[1]CAS Center for Excellence in Nanoscience, Beijing Institute of Nanoenergy and Nanosystems, Chinese Academy of Sciences, Beijing 101400, China. [2]School of Nanoscience and Technology, University of Chinese Academy of Sciences, Beijing 100049, China. [3]Key Laboratory of Advanced Materials (MOE), School of Materials Science and Engineering, Tsinghua University, Beijing 100084, China. [4]Department of Materials Science and Engineering, College of Engineering, Peking University, 100871 Beijing, China. [5]Center on Nanoenergy Research, School of Physical Science and Technology, Guangxi University, Nanning 530004, China. [6]School of Physical Science and Technology, Lanzhou University, Lanzhou 730000, China. [7]School of Advanced Materials, Shenzhen Graduate School, Peking University, Shenzhen 518055, China. [8]Yonsei Frontier Lab, Yonsei University, Seoul 03722, Republic of Korea. [9]School of Materials Science and Engineering, Georgia Institute of Technology, Atlanta, GA 30332-0245, USA. [10]These authors contributed equally: Yusen Su, Andy Berbille, Xiao-Fen Li, Jinyang Zhang. ✉e-mail: zhulaipan@binn.cas.cn; zhong.wang@mse.gatech.edu

sieving by adsorption on various substrates[8], or more rarely magnetoelectric materials[9,10].

Previously, various research groups explored the ability of dielectric materials, which have been pre-charged by solid-solid friction, to reduce metal ions in solution, i.e., the tribocatalytic reduction of metals[11,12]. Later research showed that the catalytic ability of the dielectric insulator towards metal reduction is not correlated with its ability to withdraw charges; the catalytic activity of tribocharged materials is rather favorized when the molecules that constitute the surface are characterized by a low ionization energy[13]. This phenomenon has been exploited by chemist to pattern surfaces with metal particles by rubbing two insulators together[12], or more recently to reduce $CO_2$ in presence of water by friction of copper on glass[14], or trigger atomic transfer radical polymerization[15].

Recently, we discovered that water-solid contact-electrification (SL-CE) can be used to catalyze chemical reactions[16]. This form of catalysis, called contact-electro-catalysis, takes advantage of electron exchanged by contact electrification during contact-separations cycles between a dielectric insulator and a liquid, generally an aqueous solution[17], to catalyze redox reactions[18]. Since then, researchers found the effects of CEC can be obtained in various conditions including ultrasonic[18] and microfluidic conditions[19], or by sliding droplets on a slope under illuminations[20]; CEC has also been proven to influence the outcome of some chemical reactions in microdroplets chemistry[21]. Moreover, it appeared that, contrary to tribocatalysis, the performances of dielectric insulators towards CEC at room temperature is correlated with their ability to withdraw electrons from water by contact-electrification. Since its discovery in 2022, ultrasonically assisted CEC has been successfully employed for the degradation of organic dyes[16], using powders and films[22] as catalysts, for the leaching of Li and Co from spent Li-Ion batteries cathode materials[23], and for hydrogen peroxide evolution[18]. All these processes were mostly oxidative processes, generally using polymers as catalysts at ambient temperature and in water, at the exception of the battery recycling experiment that was conducted at high temperature in presence of $SiO_2$, an acid, and at higher temperatures.

Here, we show the principles of CEC can be employed to develop a metal-free catalytic path towards the reduction of precious metals from aqueous solutions. We discovered microparticles of fluorinated ethylene propylene (FEP), owing to its high CEC activity, can drive the CEC reduction of ions of gold (Au), mercury (Hg), palladium (Pd), platinum (Pt), iridium (Ir), rhodium (Rh) and silver (Ag) from aqueous solutions, in both aerobic and anaerobic conditions. Moreover, FEP can successfully extract gold from solutions of concentrations ranging from about 0.001 to 1 mM in this experiment. These results were compared to currently published methods for gold extraction, both in terms of concentration range and extraction capacity. The effect of oxygen on the kinetic of reaction was studied through electron paramagnetic resonance spectroscopy and in ab-initio simulation.

After optimizing the conditions of the gold reduction reaction (mass loading, temperature, ionic strength, and pH), the CEC activities of various dielectric polymers towards the reduction of $AuCl_4^-$ by CEC were tested. Until now, the link between CE, CEC activity and the ejection electrons in water has not been thoroughly established for a large range of polymers[18]. To address this issue, we therefore compared the results obtained for the reduction of $AuCl_4^-$ by various catalysts to the contact-electrification ability of these materials and their ability to reduce TEMPO, an electron scavenger used in EPR to capture electrons.

Finally, we explored the possibility for FEP to perform the catalytic extraction of gold from e-waste leachate by ultrasonic CEC. Our method was successfully applied to gold extraction from leachates obtained by treating spent central processing units (CPUs) and electroplating waste.

## Results

### Reduction of metal ions in aqueous solution by ultrasonication in presence of dielectric polymer microparticles

In this experiment, as shown in Fig. 1a, a beaker containing a solution (50 mL) of metal ions and a certain amount of dielectric insulator powders (polymer or ceramics) is ultrasonicated (120 W, 40 kHz) for a given duration. To limit the potential influence of oxygen, we conducted these experiments in anaerobic conditions first. After a few minutes of ultrasonication, the solution acquires a coloration, and a precipitate forms in the reactor (Fig. 1a). This experiment was conducted using FEP, polytetrafluoroethylene (PTFE), polypropylene (PP), and high-density polyethylene (HDPE), as catalysts for the reduction of $AuCl_4^-$ in solution. HDPE, a material endowed with negligible water-solid contact-electrification ability (Fig. 1b), presents similar performances to that of the control experiment (without catalysis). Meanwhile, consistently with previous CEC experiments, FEP stands out as the best performer[16,18]. This outstanding performance results from the high electrons withdrawing ability of fluoride atoms during water-solid CE (Fig. 1c)[24]. We compared these results with the ability of the various particles to reduce 2,2,6,6-Tetramethylpiperidine 1-oxyl (TEMPO), a paramagnetic electron scavenger, by EPR spectroscopy (Fig. 1d). When TEMPO is reduced to TEMPOH it loses its paramagnetic properties, resulting in a decreasing EPR signal. It appears that the ability of the materials to reduce TEMPO follows the same trend as what was observed for the reduction of $AuCl_4^-$ (FEP > PTFE > PP > HPDE > Control, see Fig. 1a) and for the water-solid CE experiment (Fig. 1b). These observations suggest that the catalytic activity of the dielectric powders in CEC is linked to its ability to perform water-solid contact-electrification and to release those electrons. This confirms that the phenomenon at play here is indeed CEC, and not due to a solid/solid contact-electrification that is used to perform tribocatalytic reactions, as evocated in the introduction[13]. To further confirm this result, we performed a capture experiment that shows that dimethyl sulfoxide, an electron scavenger, was the most capable of hindering the reaction of gold reduction by CEC (Supplementary Fig. 1). Finally, to exclude the possibility that the decrease of gold concentration could result from a potential ultrasonic absorption of gold on FEP, we conducted an intermittent experiment (30 min US ON, 30 min US OFF) to evaluate the influence of adsorption/desorption cycles. Whether a power of 120 W or 60 W is used, we only observe a slight desorption of gold, below ppb level, after the adsorption equilibrium is reached (30 min), between each ultrasonication cycles, see Supplementary Fig. 2 and Table 1. The result of the intermittent experiment compared with that of the continuous experiment, shown in Supplementary Fig. 2, do not show a significant difference between the two setups. These results could be explained by the low surface area (9.023 $m^2 g^{-1}$) and total pore volume (0.0288 $cm^3 g^{-1}$) of the particles employed (see Supplementary Fig. 3, and Supplementary Table 1 and 2). These observations confirm that CEC is the main driving mechanism behind the extraction of gold.

Prior to further experiments with various metal ions, we optimized the conditions of the reaction by successively studying the effects of catalyst loading, temperature, ionic strength, pH, and particle size on the reduction of $AuCl_4^-$ by CEC, for 1 h (Supplementary Figs. 4 and 5). The observations are mostly consistent with previous literature reporting the performances of FEP for CEC[18,25]. The optimal catalyst loading is 1:5000 (FEP:Water) (Supplementary Fig. 4a) and the temperature is kept at 25 °C (Supplementary Fig. 4b). Because of the charge screening effect arising under high ionic strength conditions[17], which results in a decreased electron exchange by water-solid CE, adding NaCl to the solutions slows down the reaction (Supplementary Fig. 4c). Similarly, adjusting the pH of the $AuCl_4^-$ solution results in a decrease of the catalytic activity owing to the consequential increase of ion concentration (Supplementary Fig. 4d). In Supplementary Fig. 5a and b, particles of size were tested, showing that the catalytic efficiency improves as the particles size decreases from 30 to 6.5 μm; however, it only marginally

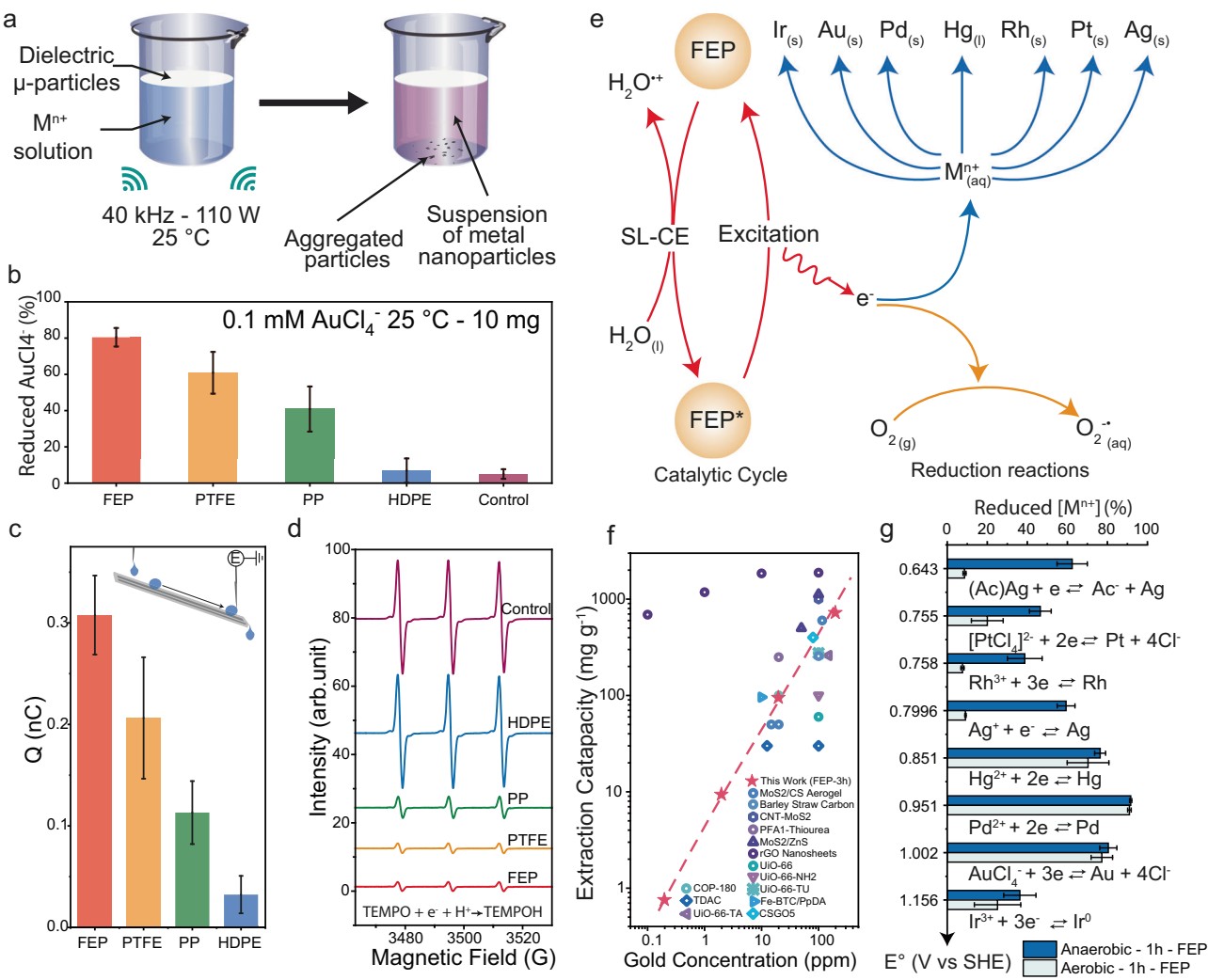

**Fig. 1 | Reduction of metal ions in solution by ultrasonication in presence of fluorinated ethylene propylene (FEP) microparticles. a** Graphical description of the experimental set-up and observations. **b** Reduction of $AuCl_4^-$ to $Au^0$ by CEC using various dielectric powders as catalysts. **c** Measurement of the charge of a microdroplet sliding on a slope made FEP, PTFE, PP or HDPE. **d** Detection of TEMPO by EPR spectroscopy for various catalysts. **e** Schematic description of the reduction of various metal ions ($M^{n+}$) in aqueous solution by ultrasonically driven CEC in presence of FEP. **f** Gold extraction capacity for FEP-based CEC compared to adsorption/reduction method using different materials[8,36–48], on a range from around 0.1 ppm to around 100 ppm. Dotted lines are guides to eyes. **g** Reduction amount of various metal ions in aqueous solution by FEP-based CEC after 1 h. Error bars represent standard deviations for 3 reproduced experiments.

increased from 6.5 to 0.2 µm. Interestingly, despite a large increase of the specific surface area and porosity from particles of 2 to 0.2 µm, see Supplementary Table 2, we did not observe a proportional increase of the gold extraction efficiency. This could be explained by the fact that, as demonstrated by a previous experiment, the surface composition is more conducive to the CEC performance of a material than its surface area[24]. The experiment was also conducted in other solvents than water, namely dimethyl sulfoxide (DMSO), Acetonitrile, or ethanol, and in the same conditions of temperature, power, concentration and catalyst loading as for water. For all these 3 solvents, we did not observe any significant decrease of the concentration of the $AuCl_4^-$ (see Supplementary Fig. 6). According to the manufacturer, the FEP resin our particles are made from does not contain additives (see Method section). However, we considered the possibility that additives leaking from other commercial polymers could influence the reaction. In Supplementary Fig. 7, the results of gold reduction experiments in presence of trace amount of additives (~$10^2$ ppb) in the solution show that additives do not affect the CEC gold reduction process.

Figure 1e contains a schematic description of the mechanism of metal ions reduction by CEC. We propose that after ultrasonically

assisted water-solid CE, then charged FEP, noted FEP*, emits electrons towards the solution under the excitation provided by ultrasonic vibrations[19]. These electrons can then reduce the metallic ions present in the solution. After the electrons left FEP*, the latter returns to its ground state (FEP), and the catalytic cycle continues if ultrasonic conditions are maintained. A more detailed explanation of the mechanism of contact-electrification and CEC is presented in Supplementary Note 1. As shown in this figure, and as described in more detail later in the manuscript, the reduction of metals by CEC can be employed to extract Ir, Au, Pd, Hg, Rh, Pt, Ag from aqueous solutions.

To evaluate the performances of FEP-based CEC as a gold extraction method, we attempted this experiment on a 50 mL synthetic $AuCl_4^-$ solutions at 0.001, 0.01, 0.1, and 1 mM. In these cases, as reported in Fig. 1f (red stars), we reached an extraction capacity after 3 hours of, respectively, 0.756, 9.387, 95.706, and 722.5 mg g$^{-1}$. This result is on par with most adsorbent-based processes presented in recent literature (Fig. 1f, Supplementary Table 3) and is applicable at very low concentrations (Supplementary Fig. 8).

In Fig. 1g, we reported the results obtained for the reduction of Ag(Ac), $Rh^{3+}$, $[PtCl_4]^{2-}$, $Ag^+$, $Hg^{2+}$, $Pd^{2+}$, $AuCl_4^-$ and $Ir^{3+}$, after 1 h. It

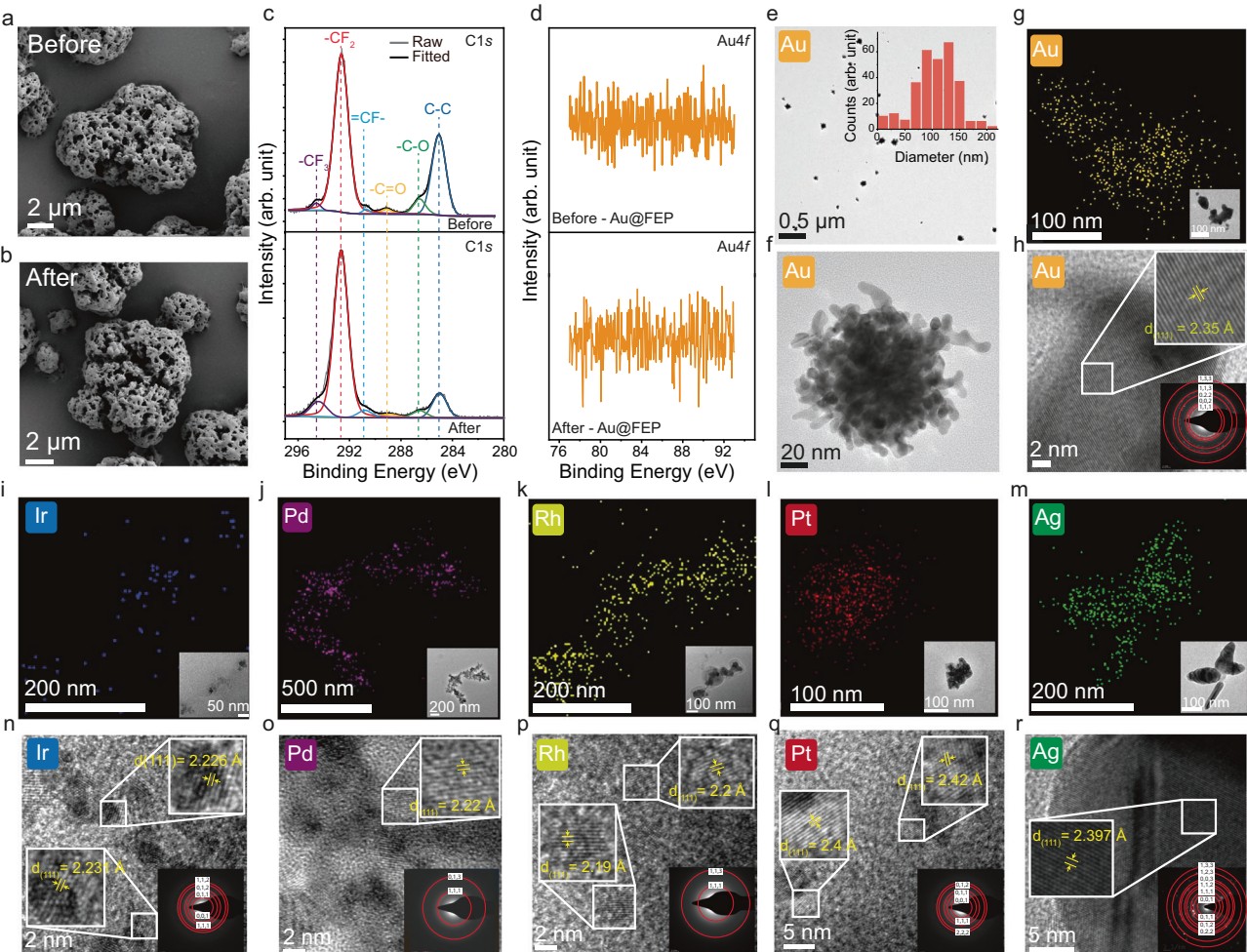

**Fig. 2 | Post-reaction characterization of the catalyst and precipitates.** SEM photographs of the FEP particles (**a**) before and (**b**) after the reaction. **c** C1s XPS spectrum of FEP before (top) and after (bottom) the reaction. **d** Au4f XPS spectrum of FEP before (top) and after (bottom) the reaction. **e** TEM photograph, and particle size distribution of Au nanoparticles. **f** TEM image of gold nanoparticle dispersed in water. **g** EDX mapping of gold; the inset is a TEM images of the sample analyzed by EDX. **h** Measurement of d-spacing on gold, the bottom right inset contains the SAED pattern of the particles. EDX mapping of (**i**) Ir, (**j**) Pd, (**k**) Rh, (**l**) Pt, (**m**) Ag. Measurement of d-spacing on (**n**) Ir, (**o**) Pd, (**p**) Rh, (**q**) Pt, (**r**) Ag; the insets contain the SAED pattern of the particles.

appeared all those ions can be reduced by CEC, but at different rates. It is worth noting there does not seem to be a link between the reduction potential and the kinetic of the reaction. Since $O_2$ tends to accumulate at the surface of fluorinated polymers in water[26], and considering the potential role of $O_3$ evocated in previous experiments[18], we compared the results obtained for all these metals in anaerobic and aerobic conditions. After 1 h, we consistently observe a higher amount of metal ions is reduced in most cases under anaerobic conditions. Only a slight difference is observed in the case of Hg, Pd, and Au.

## Characterization of the catalyst and the precipitate after reaction

To ensure the CEC reduction of metals is a catalytic process we studied the physicochemical properties of FEP before and after the reaction. Scanning electron microscopy (SEM) photographs and EDX of the FEP microparticles do not demonstrate significant changes after the experiment (Fig. 2a, b, Supplementary Fig. 9 and 10). The particle size analysis presented in Supplementary Fig. 11 shows a negligible decrease of their diameter after the reaction. The study of the composition FEP's surface by X-ray photoelectron spectroscopy (XPS) shows the chemical composition of the surface did not change drastically after the experiment (Fig. 2c and Supplementary Fig. 12). The intensity of the peak for the binding energy of C-C on FEP in the C1s spectra decreased, which could be related to the slight decrease particle size mentioned earlier[22]. Additionally, we examined the Au4f spectra at the surface of FEP and did not find bonds between FEP and gold atoms (Fig. 2d). Similar conclusions are drawn from Fourier transformed infrared spectroscopy (Supplementary Fig. 13). These results show that FEP is a robust catalyst that could have a good recyclability, owing to its high chemical and mechanical stability.

The solid precipitates obtained at the end of the gold reduction experiment were examined by high-resolution transmission electron microscope (HRTEM). The diameter of the particles obtained by the reduction of $AuCl_4^-$ via CEC ranges from 100 nm to 150 nm (Fig. 2e) when dispersed in water. In this case, the particle adopts a spherical shape with aggregates (Fig. 2f). However, the particle unfolds when dispersed in alcohol, revealing a structure made of nanowires and spherical particles (Supplementary Fig. 14). Using EDX, we could confirm that the particles contained gold element (Fig. 2g and Supplementary Fig. 15). The composition of the particle is confirmed by the data obtained in XPS (Supplementary Fig. 16). We then used SAED and image analysis to obtain the Miller indices (hkl) and d-spacing (Fig. 2h) of one of the particles. We measured a d-spacing of 2.35 Å on (111) plane, which matches well with the data reported in the literature (PDF#04-0784). These results, combined with the study of the

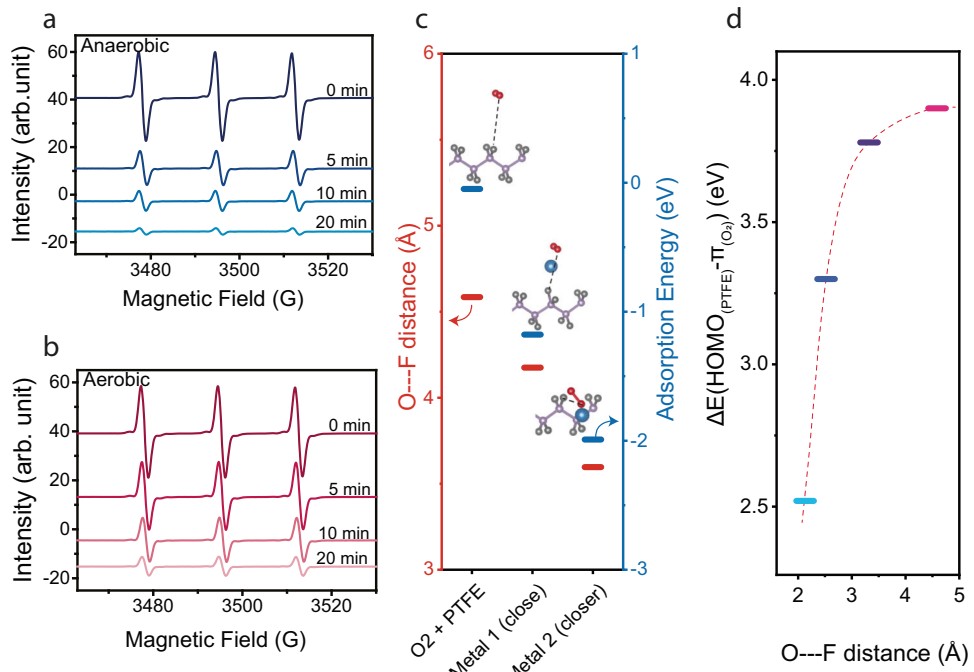

**Fig. 3 | About the influence of oxygen.** EPR signals acquired for 20 min, during the reduction of TEMPO (paramagnetic) to TEMPOH (non-paramagnetic) by ultra-sonication of a TEMPO solution in presence of FEP in (**a**) anaerobic or (**b**) aerobic conditions. **c** Calculated distance between O of oxygen and F, and adsorption energy for PTFE and $O_2$. **d** Calculated energy difference between the HOMO level of PTFE and the π-bonds levels of dioxygen (color code of atoms: Gray, Violet, Red, Blue for F, C, O, Pd(II), respectively). Dotted lines are guide to eyes.

catalyst's surface, showed that gold ions in solution have been effectively reduced to solid particles of $Au^0$.

We conducted the same characterizations using EDX, HRTEM, SAED for the product of the reduction $Ir^{3+}$, $Pd^{2+}$, $Rh^{3+}$, $PtCl_4^-$, $Pd^{4+}$ and $Ag^+$ (Fig. 2i–r and Supplementary Fig. 17). These results show all those ions have been reduced to their metallic form. The results obtained for d-spacing of each particle by analysis of the HRTEM and SAED results are presented in Supplementary Table 4.

### About the influence of oxygen

The series of experiments presented in Fig. 1g demonstrated the adverse effect of aerobic conditions on the ability of FEP to reduce metal ions by CEC. We now combine experimental and theoretical data to understand how oxygen may affect the reduction reaction.

First, we measured the discrepancy in FEP's ability to generate electrons in solution by CEC in anaerobic (Fig. 3a) and aerobic (Fig. 3b) conditions, using a similar EPR method as that presented in Fig. 1c. As expected, in the presence of oxygen, the rate at which TEMPO (paramagnetic) is reduced to TEMPOH (non-paramagnetic) by CEC is lower compared to that in anaerobic conditions. This result indicates that in aerobic conditions less electrons are available to react with species present in the solution. This could result from the effect of oxidative atmospheres on the highest occupied surface state level of dielectric polymers, which conducts to a lessened ability to gather electrons by contact-electrification[27]. However, we believe other factors might play a role, such as a favorable adsorption energy of oxygen on fluorinated polymer[26] and the ability of metal salt to influence the distance between the catalyst and the oxygen, and thus the energy necessary to transfer 1 electron from a polymer to oxygen.

To simplify our calculations, we used a short PTFE polymer chain as a catalyst in our model and palladium as the ion. Using fluorinated polymers as catalysts, in concert with oxygen–both characterized by their non-polarity–introduces a unique challenge. Indeed, the inherent resistance of these molecules to dissolve or interact with water lead to the formation of a gas barrier between the solution and the catalyst[26].

This barrier may obstruct electron transfers towards the metal ions and impede the ability of the catalyst to perform CEC reduction reactions other than that of oxygen. In Fig. 3c, we studied how the presence of metals ions, Pd here, influences the distance between molecular oxygen and fluoride atoms on PTFE. The calculation in absence of metal ions shows the F–O distance ($d_{F\cdots O}$) is 4.5851 Å, while the adsorption energy ($E_{ads}$) equals −0.0483 eV. We then introduced a Pd ion in the system at two different positions relative to PTFE, pos. 1 (close) and pos. 2 (closer), see Fig. 3c. As the metal ion gets closer to PTFE, $d_{F\cdots O}$ contracts, from 4.5851 Å (no Pd) to 4.1752 Å (pos. 1), and ultimately 3.5966 Å (pos. 2). This spatial convergence is mirrored in the adsorption energy that decreases from −0.0483 eV (no Pd) to −1.7757 eV (pos. 1), and −1.9908 eV (pos.2). Similar calculations have been performed for the case of gold, and they show similar trends in both the variations of bond length and adsorption energy (see Supplementary Fig. 18).

Finally, we studied the effect of the $d_{F\cdots O}$, on the energy difference between the HOMO level of PTFE and the energy level of the dioxygen's π-bond, $\Delta E(HOMO_{(PTFE)}-\pi_{(O2)})$. The results in Fig. 3d show that as $d_{F\cdots O}$ diminishes, $\Delta E(HOMO_{(PTFE)}-\pi_{(O2)})$ decreases from 3.90 eV at d = 4.98 Å down to 2.52 eV at d = 2.13 Å. This means that, as the distance between oxygen and PTFE decreases, electrons transfer from the catalyst to $O_2$ is facilitated by a lowered energy barrier and greater affinity between the two molecules. These factors may participate in the hindered ability of FEP to reduce some metals in aerobic condition, owing to an enhanced adsorption of oxygen at the surface of the catalyst in presence of metal ions in solution. What's more, using polypropylene as a model we found that tacticity does not have a significant influence on the reduction process by CEC. At a fixed bond length of 2.11 Å, $\Delta E(HOMO_{(PTFE)}-\pi_{(O2)})$ of isotactic, syndiotactic and atactic PP reach 1.83, 1.83, and 1.82 eV, respectively (see Supplementary Fig. 19).

### Selective extraction of Gold by CEC from CPU leachates

E-waste, such as CPUs, and electroplating waste, contains a variety of metals, among which gold is the most valuable to recycle. We found out we could take advantage of the limitations of CEC in aerobic

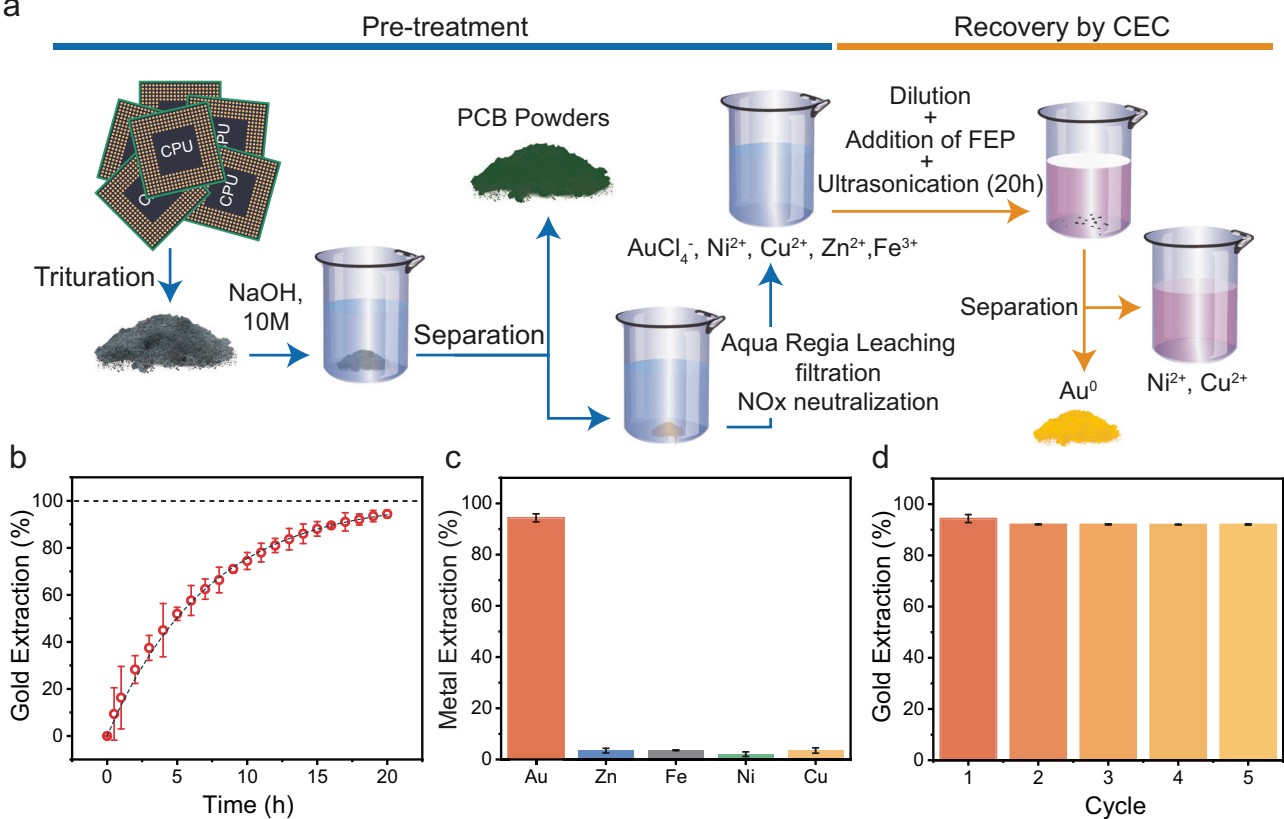

**Fig. 4 | Selective extraction of Gold from the leachate of spent CPUs by CEC.**
**a** Schematic of the pre-treatment of CPU (blue) and recovery of CEC for extracting gold from spent CPUs. Arrows are guide to eyes. **b** Evolution of gold extraction over time from a leachate of CPU at a concentration of 10 ppm of gold. The doted lines are guides to the eyes. **c** Amount of $AuCl_4^-$, $Zn^{2+}$, $Fe^{2+}$, $Ni^{2+}$, and $Cu^{2+}$ extracted after 20 h. **d** Recycling of FEP particle for 5 cycles of 18 h. Error bars represent standard deviations for 3 reproduced experiments.

condition to selectively extract Au from e-waste leachates. We firstly leached the metals from crushed CPU or electroplating wastes, as reported in Fig. 4a and Methods. They are first soaked in a 10 M NaOH solution for 48 hours to remove organic materials, and then filtered (blue arrows). Thereafter, aqua regia was employed to leach the metals (red arrows) from the solid obtained in the previous step. The solution is finally filtered, and nitrous oxides are neutralized. The solution is then diluted until Au concentration reaches ~10 ppm, a concentration often encountered in e-waste recycling[8], before starting the metal reduction by CEC in an aerobic environment (yellow arrows). The result presented in Fig. 4b and c show 94.4% of $AuCl_4^-$ has been reduced to $Au^0$ after 20 hours, while the concentration of the other ions (Zn, Fe, Cu, Ni) decreased only slightly. This shows a good selectivity of the CEC process. Moreover, the product can be separated by a simple decantation from the catalyst, since FEP floats at the surface. A similar process was conducted for electroplating waste. In this case, 91.5% of the gold in solution was extracted (Supplementary Fig. 20). This selectivity could result from a kinetic mechanism of selectivity, owing to the fact the solution was neutralized to avoid redissolution of the metals, i.e., reversible reactions[28]. To ensure it was the case, prepared individual solutions of each of the metal ions present in the leachate as individual solutions in DI water; the solutions are then ultrasonicated (40 kHz, 120 W) in presence of 10 mg of FEP, at 25 °C. We observed that, even when they are reduced in separate solution, while a great portion of the gold ions are reduced, Fe, Ni, Cu, Zn ions are barely removed from the solution after 1 hour of treatment (see Supplementary Fig. 21).

Finally, we evaluated the recyclability of FEP as a catalyst for gold extraction from e-waste water. We found no obvious decrease of the performance of FEP for gold extraction after running 5 cycles (18 h per cycle) with the same catalyst (Fig. 4d). Owing to the low price of FEP ( ~ 0.05 \$ $g^{-1}$), the simple separation of the product from the catalyst, and the ability of this method to extract gold from both low and high concentration solutions (see Fig. 1f), this method could be profitable in large scale setup if the inherent disadvantages of ultrasonic processes are addressed in the design of the processing plant[29].

## Discussion

Here, we demonstrated the ability of contact-electro-catalysis, the catalysis by water-solid CE, to perform the mechano-catalytic reduction of precious metals from aqueous solution at room temperature and pressure. Using ultrasounds in presence of FEP microparticles, Ir, Au, Pd, Hg, Rh, Pt, Ag nanoparticles were extracted from aqueous solutions, in both aerobic and anaerobic conditions. Gold can be recovered from solutions with concentrations as low as 0.001 mM. For some metal reduction reactions, we observed a stark difference between aerobic and anaerobic conditions. This could be explained by the adsorption energy of oxygen, and the reduced distance and energy barrier between the catalysts and the oxygen which are further reduced in presence of Pd. Consequently, many electrons can be gathered by the highly oxidative oxygen rather than by the target of the reaction. This information should be considered in the design of future experiments. Moreover, we showed gold can be extracted by CEC methods from the leachate of CPUs and electroplating waste. This method, endowed with good selectivity, relies on highly recyclable metal-free catalysts. This work also allowed us to propose a better demonstration of the relationship between the CE ability of a polymer and its ability to perform CEC. We showed a correlation between the CEC activity of FEP, PTFE, PP, and HDPE, their ability to generate electrons in solution during a CE experiment and their CE performances.

As a method to extract metals from aqueous solution, CEC holds several advantages. The commercial availability of microparticles made from readily available high-performance dielectric insulating fluoropolymers and ceramics, removes the obstacle that usually represents the scaling of novelty catalyst and sorbents. Because fluoropolymers are chemically inert, rather than being immobilized on the catalyst, floating at the surface, the extracted metal precipitates at the bottom of the reactor, which facilitate its recovery compared to sorbent processes that requires further post treatments, namely selective dissolution and reprecipitation of the compound of interest; CEC is a one-step process. Furthermore, the CEC reduction of metals allows for the recovery of metals from solutions of low pH, such as industrial leachate obtain from electroplating waste or discarded CPU. Owing to the great difference in kinetic between the reduction of gold and other metals, CEC allows for a selective recovery of gold from a leachate of those same leachatese-wastes. Finally, the physicochemical properties of FEP allowed that material to display an excellent recyclability.

Furthermore, the CEC reduction of metals allows for the recovery of metals from solutions of low pH, such as industrial leachate. Owing to the great difference in kinetic between the reduction of gold and other metals, CEC allows for a selective recovery of gold from a leachate of those same leachates. Finally, the physicochemical properties of FEP allowed that material to display an excellent recyclability. We believe future works should focus on improving the efficiency of CEC processes, and finding new materials which are less polluting than fluorinated polymers and continue to explore the range of reactions that can be performed by this method. The potential contribution of CEC to other forms of catalysis, and mechanochemical processes is also something that should be considered and explored.

## Methods

### Reagents and materials

Fluorinated ethylene propylene (Teflon FEP 100, Dupont, Mn = 676 000 000), Polytetrafluoroethylene (Teflon PTFE 7AX, Dupont, Mn = 316 000 000), Polypropylene (PP, Macklin, Mw = 289 300, (see Supplementary Fig. 22)), High-density polyethylene (HDPE, Sigma-Aldrich, Mw = 3 000 0000 ~ to 6 000 000), Silicon dioxide ($SiO_2$, Aladdin, 99.95%), p-Benzoquinone ($C_6H_4O_2$, Macklin, ≥99.5%), Dimethyl sulfoxide ($C_2H_6SO$, Macklin, 99.8%), Ethylenediaminetetraacetic acid disodium salt ($C_{10}H_{14}N_2$ $Na_2O_8 \cdot 2H_2O$, Aladdin, 99%), Tert-butanol ($C_4H_{10}O$, Macklin, 98.0%), Silver nitrate ($AgNO_3$, Macklin, 99.8%), Rhodium(III) nitrate dihydrate ($H_4N_3O_{11}Rh$, Macklin, 95%), Iridium chloride(III) hydrate ($IrCl_3 \cdot 3H_2O$, Macklin, Ir>52%), Mercury nitrate monohydrate ($HgN_2O_6 \cdot H_2O$, Macklin, 99%), Silver acetate ($C_2H_3O_2Ag$, Macklin, 99.5%), Palladium nitrate dihydrate ($Pd(NO_3)_2 \cdot 2H_2O$, Macklin, Pd≥39%), Potassium gold chloride ($KAuCl_4$, Macklin, Au≥51%), Potassium(II) tetrachloroplatinate ($K_2PtCl_4$, Macklin, 98%), 5,5-dimethyl-1-pyrroline N-oxide [$C_2H_6OS$, Dojindo], 2,2,6,6-tetramethylpiperidinooxy (TEMPO, $C_9H_{18}NO$, Macklin, 98%), Sodium chloride (NaCl, Macklin, 99.5%), Hydrochloric acid (HCl, Macklin, AR), Nitric acid ($HNO_3$, Macklin, AR), Sodium hydroxide (NaOH, Macklin, 99.5%), DMSO (Macklin, 99.7% Water ≤ 50 ppm), Ethanol (Macklin, 99.7%), Acetonitrile (Sinopharm Chemical Reagent Co. Ltd., 99%). CPUs, plating wastes, and dismantled factory waste machines were purchased from Alibaba.

The FEP, PTFE, PP and HDPE were characterized by Fourier Transformed Infra-Red Spectroscopy (FTIR), Thermogravimetric Analysis-differential scanning calorimetry (TG-DSC), and Gel Permeation Chromatography (GPC) (See Sample Characterization and Supplementary Figs. 22–24).

### Metal reduction using contact-electro-catalysis

The catalyst (10 mg) is introduced into a gas washing bottle (see Supplementary Fig. 25) containing a solution (50 mL) at a certain concentration of metal ions (0.001 mM, 0.01 mM, 0.1 mM, or 1 mM). The reaction system is then ultrasonicated (120 W, 40 kHz) for a given amount of time (1 h, 3 h, or 20 h). The temperature of ultrasonic water was controlled by a thermostat connected to a radiative copper cooler placed in the ultrasonic bath at 25 °C, unless otherwise specified.

In aerobic conditions, the protocol is followed without needing any further preparation.

Anaerobic conditions are created by saturating the solution with gaseous $N_2$ gas (15 min). The inlet and outlet valves of the sealed gas washing bottle are then closed. The reactor is finally placed into an ultrasonic bath (40 kHz, 120 W). The solution is saturated again with $N_2$ after each aliquot (1 mL) has been sampled.

The concentration of metal ions was measured by inductively coupled plasma optical emission spectroscopy (ICP-OES) and inductively coupled plasma mass spectrometry (ICP-MS).

### Reduction rate and extraction capacity

The reduction efficiency of each reaction after a given amount of time, R in %, is calculated as:

$$R = 100 * \frac{(C_0 - C_e)}{C_0} \tag{1}$$

where $C_0$ is the initial concentration of $PM^{n+}$, $C_e$ is the final concentration after the reaction.

The extraction capacity is calculated as:

$$Q = \frac{(C_0 - C_e) * V}{m_{cat}} \tag{2}$$

where $V$ is the volume of reaction solution and $m_{cat}$ is the mass of the catalyst.

### Extraction of gold from e-waste leachate

Two kinds of e-waste for real-world application: central processing unit (CPU) and electroplating waste.

The CPUs are first crushed to a powder that is soaked in a 10 M NaOH solution for 48 hours to remove printed circuit board on the surface. After separation, the CPUs were rinsed and soaked in 500 ml aqua regia (HCl: $HNO_3$ = 1:3) until all metal from the surface are dissolved in the solution. The concentration of gold in the solution reaches about 100 mg $L^{-1}$ at that stage. The solution is then filtered. Nitrous oxides are neutralized by evaporating the solution until no red colored gases are emanated from the solution, while HCl is dropwise added to the solution. The solution is diluted 10 times, with a pH remaining at -1.35. The pH is not adjusted further, and 10 mg of FEP powder is added to 50 mL of the as obtained solution. The reactor is then ultrasonicated in aerobic conditions.

Electroplating waste is directly treated with aqua regia, without any pre-treatment, until the coating is fully dissolved. All subsequent steps are the same as previously described CPUs.

The amount of gold extracted is evaluated by ICP-MS and using the formulas (1, 2).

### Capture experiments

The contribution of various species to the reduction reaction were investigated by a series of capture experiments (1 h). Various trapping agents (0.1 mM) are added into a solution of $AuCl_4^-$ (0.1 mM), namely Ter-butanol, p-benzoquinone, DMSO, and EDTA-2Na, capturing $\cdot OH$, $O_2^{\cdot -}$, electrons, and holes, respectively.

### Electron Paramagnetic Resonance Spectroscopy

TEMPO was employed to detect the activity of the various particles, in anaerobic or aerobic conditions as well as for FEP, with regards to the production of electrons. TEMPO is a stable radical at ambient temperature that reacts with 1 electron and 1 proton in solution to form the non-paramagnetic TEMPOH. Measuring a rapidly decaying signal of TEMPO is conducive to its reduction by the catalyst.

Prior to ultrasonication, 62.5 μL of a TEMPO solution (25 mM) is dissolved into 10 ml DI water to which 10 mg catalyst is added. The signal of TEMPO is then measured at given time intervals during ultrasonication on a Bruker EMXplus-9.5/12. The measurements were conducted in X-Band (9.813926 GHz), with amplitude modulation of 2 G, microwave power of 20 mW, and an amplitude modulation frequency of 100.00 kHz and conversion time of 26.70 ms, and a time constant at 0.01 ms.

**Measurement of single droplet TENG.** The studied dielectric films are adhered to a PMMA slope placed at an angle of 50° relative to ground. Before each measurement, the surface of the polymer is cleaned with Milli-Q water, ethanol and blown over with an ion fan for 10 min to ensure no impurities or charges remain on the surface. A 40 μl droplet of Milli-Q water is released 5 cm above the sample by a grounded syringe needle (14 G), by actuating a syringe pump (WH-SP-04, Wenjing). A needle electrode placed close to, but not touching, the film's surface collects the charge of the droplet at the end of its course (5 cm). The signal is acquired through an electrometer (Keithley 6517B). The error bars are obtained for three repeatable experiments.

**Sample characterization.** ICP-OES measurements were performed on an Agilent ICP-OES 730, while ICP-MS measurements were conducted on an Agilent ICP-MS 7850.

SEM characterization was conducted on a Zeiss sigma 300, or on Zeiss Merlin Compact.

FTIR spectroscopy was conducted on Bruker VERTEX80v or Thermo Fisher Nicolet iS 10 on a range from 400 to 3000 cm$^{-1}$.

XPS characterization were conducted on a Kratos AXIS Ultra DLD using Alka-ray source (hv= 1486.6 eV). Operation vacuum, voltage, filament current, and pass energy is $1 \times 10^{-9}$ mBar, 15 kV, 10 mA and 30 eV.

HRTEM photographs and EDX mapping and spectra were obtained on a Tecnai G20 20 TWIN UEM.

The specific surface area and pore size distribution of FEP powder were obtained using the Brunauer– Emmett–Teller (BET) approach with BSD-660S.

The TG-DSC analysis was conducted on NETZSCH STA 449 F5 using N$_2$ atmosphere. The ramp is 10.00 °C per min.

GPC was employed to measure the molecular weight of PP. The experiment was carried on Agilent PL-GPC 220 with PLgel 10um MIXED-B LS 300×7.5 mm tandem column, using 150°C 1,2,4-Trichlorobenzene as the solvent. The flow is 1 mL min$^{-1}$.

The molecular weight of FEP and PTFE was calculated based on the data provided by Dupont (SSG method), using the formula[30]:

$$Log\overline{Mn} = \frac{2.61 - SSG}{0.06} \qquad (3)$$

**Calculation methods.** The density functional theory calculations were implemented in the Vienna ab initio Simulation Pack (VASP 6.2.1) code[31]. Projector-augmented wave (PAW) approach was used to describe electron-ion interactions[32]. The generalized gradient approximation of Perdew-Burke-Ernzerhof (PBE) were employed for geometry optimization[33] and self-consistent static calculations[34], respectively. The kinetic energy cutoff was set to 520 eV. A conjugate gradient method was applied for geometry optimization, with a Gaussian smearing width of 0.05 eV. The total energy criterion in the electronic self-consistency loop and the force criteria in the ionic relaxation loop was set to $10^{-4}$ eV and 0.05 eV Å$^{-1}$, respectively, when calculating the adsorption energy and difference of activation energy in combination with a Γ-centered 2×2×10 k-mesh[35].

## Data availability

The data supporting the findings of this study are reported in the main text or the Supplementary Information. Raw data can be obtained from the corresponding authors upon request. The atom coordinates for the DFT calculations are available from Materials Cloud at the address: ref. 48.

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

## Acknowledgements

We thank Dr. Ke Tian from institute of process engineering (CAS) for her technical assistance on EPR. This research was supported by the National Natural Science Foundation of China Grants 22102010 (J.Z), 52192610 (L.Z) and 52192613 (Z.L.W), National Key R & D Project from Minister of Science and Technology 2021YFA1201601 (Z.L.W) and sponsored by the CAS-TWAS President's Fellowship (A.B).

## Author contributions

Y.S, A.B, X-F. L., and J.Z contributed equally to this work. L.Z., and Z.L.W. acquired fundings and supervised the experiment. Y.S., A.B., X-F. L., and J.Z. analyzed the results and prepared the manuscript. M.J.P. performed SEM measurements. H.L., Z.L., S.L. and J. L. assisted with collecting the data. All the authors discussed the results and commented on the manuscript.

## Competing interests

The authors declare no competing interests.
