## [Peer Review File · Nature Communications]

Reduction of precious metal ions in aqueous solutions by contact-electro-catalysisREVIEWER COMMENTS

Reviewer #1 (Remarks to the Author):

In this manuscript, CEC was employed to reduce the ions of precious metals in both anaerobic and aerobic conditions. The use of fluorinated ethylene propylene microparticles can drive the CEC reduction of ions of gold (Au), mercury (Hg), palladium (Pd), platinum (Pt), iridium (Ir), rhodium (Rh) and silver (Ag) from aqueous solutions under ultrasonication. This method is capable of extracting gold from synthetic solutions with concentrations ranging from as low as 0.196 ppm up to 196 ppm, reaching in 3h extraction capacities ranging from 0.756 to 722.5 mg/g in 3 h. Finally, the authors showed CEC can be employed to design a metal-free, selective, and recyclable catalytic gold extraction methods from e-waste aqueous leachates. However, this work lacks the unique merits that suitable for Nature Communications.

1. The original CEC concept has been proposed, and the related paper has been published Nature Communications by the authors. This work is only a specific application of CEC but didn't show convincing advantage of CEC in gold extraction over other published papers as shown in this manuscript.
2. Basically, CEC belongs to mechanochemistry. Similar work related to mechanochemical reduction of metal ions have been reported (J. Am. Chem. Soc. 2015, 137, 5, 1726–1729, Applied Materials Today 26, 101339, Mater. Horiz., 2016,3, 113-118).

If the authors can understand, I would like to ask them to consider the following points.

1. The description in the main text "CEC has been successfully employed for the degradation of organic dyes, the leaching of Li and Co from spent Li-Ion batteries cathode materials, and hydrogen peroxide evolution" is not comprehensive. Actually, CEC has been also employed in the fabrication of flammable gases and controlled radical polymerization...
2. Essentially CEC process is a heterogeneous catalytic process. The adsorption-desorption equilibrium is very important. What's the effect of ultrasound on this equilibrium?

Reviewer #2 (Remarks to the Author):

This manuscript reported a metal free mechano-catalytic methods to efficiently degrade refractory organic compounds. The authors also designed a metal-free, selective, and recyclable catalytic gold extraction methods from e-waste aqueous leachates. This work is interesting. Still, there are some issues need to be addressed before the consideration of publication:

1. The authors discussed the recyclability of the FEP catalyst and drew the conclusion for only one work cycle. More work cycles need to be conducted and also long-term stability of this catalyst need to be tested.
2. The energy conversion efficiency of this CEC catalysis should be calculated and compared to the traditional methods.

3. The catalysts used in this work, like FEP, PTFE, PP, and HDPE, are all hydrophobic. When they were soaked in aqueous solution, is there any chance for them to bring air in the reaction. Will the air be involved in the reduction?
4. This work only presented the catalysis in aqueous solution. How about other solvent?
5. As we know, the triboelectric induced charge density is highly related to the surface property between two friction layers more than the intrinsic property. In this case, please discuss the effects of the factors, such as the particle size and the porosity of FEP, to the catalysis process.
6. If the reduction happened in multi-ions high conductive solution, like 0.1~1 M NaCl solution, will the CEC catalysis still be effective?
7. Please explain the working mechanism of the selective reduction of Au in the CPU leachates.

We warmly welcome the reviewers' comments and deeply appreciate the fact they have taken the time to examine the manuscript in great detail and provided many pertinent comments. The manuscript has been greatly improved following this round of review; we believe we have addressed the comments and concerns of the reviewers appropriately.

REVIEWER COMMENTS

Reviewer #1 (Remarks to the Author):

In this manuscript, CEC was employed to reduce the ions of precious metals in both anaerobic and aerobic conditions. The use of fluorinated ethylene propylene microparticles can drive the CEC reduction of ions of gold (Au), mercury (Hg), palladium (Pd), platinum (Pt), iridium (Ir), rhodium (Rh) and silver (Ag) from aqueous solutions under ultrasonication. This method is capable of extracting gold from synthetic solutions with concentrations ranging from as low as 0.196 ppm up to 196 ppm, reaching in 3h extraction capacities ranging from 0.756 to 722.5 mg/g in 3 h. Finally, the authors showed CEC can be employed to design a metal-free, selective, and recyclable catalytic gold extraction methods from e-waste aqueous leachates. However, this work lacks the unique merits that suitable for Nature Communications.

We thank Reviewer #1 for examining our paper and providing their feedbacks. We present below a point-by-point response to the comments and questions that were raised. We are confident we addressed the concerns of Reviewer #1 appropriately and improved the manuscript based on their comments.

Comment 1. The original CEC concept has been proposed, and the related paper has been published in Nature Communications by the authors. This work is only a specific application of CEC but didn't show convincing advantage of CEC in gold extraction over other published papers as shown in this manuscript.

Response: Thank you for your insightful comments and for highlighting that we did not properly present the novelty of our current work in the first version of the manuscript. We welcome the opportunity to shed light on different aspects and contributions of this research in order to help highlight how the present manuscript represents a conceptual advance; we then corrected the manuscript to reflect this process better.

It should be noted that CEC is an emerging area. For a field to grow, more researchers must be involved. Therefore, the related works of CEC should be done for many times. Up to date, there are several works about CEC that have been reported. These works either have major breakthroughs in principle or have new applications. The importance and novelty of these works may vary, but they all have unique meaning. Indeed, any progress in materials, principles or applications related to CEC is a contribution to the field. However, it doesn't mean that the researches in this field are saturated. There are still a lot of important issues needed to be addressed.

First, we would like to explicitly mention the novelty of the current work does not lie in the reporting of CEC itself, which was the purpose of our paper published, 2 years ago, in 2022 (*Nat Commun* **13**, 130 (2022)). In that first publication, we demonstrated that ultrasonically water in presence of microparticles of dielectric insulators generates reactive oxygen species (ROS) in an amount that is sufficient for degrading organic dyes, an oxidative process. In *Nat Commun* **13**, 130 (2022), we attributed this catalytic phenomenon to the charge exchange occurring by contact-electrification at water-solid interface as driven by the high frequency contact-separation cycles the birth and collapse of cavitation bubbles in ultrasonic conditions induces. The hypothesis according to which simple contact electrification between water and solid without pre-charging of the tribo-materials can drive chemical reactions, i.e., contact-electro-catalysis, has been then further verified in absence of ultrasounds in the case of *Water Research* **226**, 119242 (2022), *PNAS* **119**(32), e2209056119 (2022) (the latter proposing a secondary path, on surfaces rich in hydroxyl functional groups), and under ultrasonic conditions in the case of *Nano Energy* **112**, 108464 (2023).

In the current paper, we employed several fluorinated and non-fluorinated polymers (dielectric insulators) to reduce metals in solutions by ultrasonically assisted contact-electro-catalysis; this phenomenon has not been previously reported. Many papers are published each year about precious metal ions capture/reduction by photocatalysis, sorbent-based process, electrochemical fuel production, and others, which are much older concepts, in a wide range of journals, including high impact journals. We do not think the fact that 2 different experiences are based on the same phenomenon, sometimes by the same authors, deem them inappropriate for publication in the same journal; we fail to understand why CEC should be treated differently in that regard.

In the present paper, apart from the demonstration of a reaction that has not been previously reported for CEC (reduction of metal ions by the charge exchange brought up by ultrasonically assisted contact electrification at water-solid interface), we show that:

1. CEC can conduct reduction reactions of metal ions by in-situ generation of electrons in water through the sole contact-electrification at water-solid that is induced by ultrasonic cavitations. To the best of our knowledge, the reduction of metal by CEC has not yet been reported in the literature.
2. We provide evidence the ability of polymers towards the in-situ generation of electrons in an aqueous solution, and reduce metal ions, during ultrasonication is related to their ability to withdraw charges from the water during contact-electrification. This demonstration cements previous hypotheses proposed in *Nat Commun* **13**, 130 (2022) and concerning the nature of contact-electro-catalysis (catalysis driven by contact-electrification at solid-liquid interfaces).
3. CEC expands the range of methods available for chemists to conduct reduction reactions, and that of materials to design heterogenous catalysts from metals and metal oxides to dielectric insulating polymers.

To clarify the difference between our previous papers and the present one, we made an illustration that you will find below.

Figure R1. Illustration of the difference between this work and previous works.

Secondly, we want to mention that CEC does hold significant advantages over the other published papers.

1. For CEC, the catalyst is a microparticle, with a simple design. Despite that, the process matches the performances of most sorbent-based experiments for gold recovery (See Figure 1).
2. CEC is a form of heterogeneous catalysis capable of extracting gold from solutions with very low concentrations, which is seldom reported in the literature. Indeed, the only reported method based on heterogeneous catalysts we found capable of operating at concentrations as low as ours (industrial leachate) is *Li, F., Zhu, J., Sun, P. et al. Highly efficient and selective extraction of gold by reduced graphene oxide. Nat Commun* 13, 4472 (2022). This is of interest for practical applications, as those last moles of gold are extremely difficult to recover (see the lack of papers exploring this range).
3. CEC method for gold recovery from CPU leachate is highly selective, which is rarely encountered in the literature.
4. Moreover, in our case, the recovery of metals from the solution is a one-step process that includes both capture and reduction of the ions at once; the metal particles are not formed at the surface of the catalyst, but as free particles that precipitate at the bottom of the reactor while the catalyst floats at the water's surface. This is a significant advantage compared to the above-mentioned absorbent-based methods; CEC does not

require a subsequent leaching of the metal of interest (gold here) from the recovery system, followed by a reprecipitation to obtain the metal. 2 steps are saved.

5. CEC is endowed with long-time stability and excellent recyclability. This is owed to the chemical inertness of fluorinated polymers and their resistance to abrasion (see figure 4 b, c, d).
6. CEC is a metal-free form of heterogeneous catalyst that does not require the use of nanomaterials, graphene or other dispendious carbon-based materials, but is rather based on widely available polymer materials.

These are the advantages of CEC over many methods currently developed. The mere fact that the gold can be recovered by simple decantation is, in our humble opinion a significant advantage of CEC.

To take into account your remark and convey better the advancement that represent the current work and the advantages of CEC over other reported with method, **we amended the manuscript as follow (changes in red below, highlighted in yellow in the manuscript):**

Conclusion:

“We showed a correlation between the CEC activity of FEP, PTFE, PP, and HDPE, their ability to generate electrons in solution during a CEC experiment and their CE performances.

As a method to extract metals from aqueous solution, CEC holds several advantages. The commercial availability of microparticles made from readily available high-performance dielectric insulating fluoropolymers and ceramics, removes the obstacle that usually represents the scaling of novelty catalyst and sorbents. Because fluoropolymers are chemically inert, rather than being immobilized on the catalyst, floating at the surface, the extracted metal precipitates at the bottom of the reactor, which facilitate its recovery compared to sorbent processes that requires further post treatments, namely selective dissolution and reprecipitation of the compound of interest; CEC is a one-step process. Furthermore, the CEC reduction of metals allows for the recovery of metals from solutions of low pH, such as industrial leachate obtain from electroplating waste or discarded CPU. Owing to the great difference in kinetic between the reduction of gold and other metals, CEC allows for a selective recovery of gold from a leachate of those same e-wastes. Finally, the physicochemical properties of FEP allowed that material to display an excellent recyclability.

We believe future works should focus on improving the efficiency of CEC processes, and finding new materials which are less polluting than fluorinated polymers and continue to explore the range of reactions that can be performed by this method. The potential contribution of CEC to other forms of catalysis, and mechanochemical processes is also something that should be considered and explored”

Introduction:

“Since its discovery in 2022, ultrasonically assisted CEC has been successfully employed for the degradation of organic dyes¹⁶, using powders and films²² as catalysts, for the leaching of Li and Co from spent Li-Ion batteries cathode materials²³, and for hydrogen peroxide evolution¹⁸. All these processes were mostly oxidative processes, generally using polymers as catalysts at ambient temperature and in water, at the exception of the battery recycling experiment that was conducted at high temperature in presence of SiO₂, an acid, and at higher temperatures.

Here, we show the principles of CEC can be employed to develop a metal-free catalytic path towards the reduction of precious metals from aqueous solutions.”

- [R1] Wang, Z., Berbille, A., Feng, Y., Li, S., Zhu, L., Tang, W. & Wang, Z. L. Contact-electro-catalysis for the degradation of organic pollutants using pristine dielectric powders. *Nat. Commun.* 13, 130 (2022).
- [R2] Song, W.-Z., Zhang, M., Qiu, H.-J., Li, C.-L., Chen, T., Jiang, L.-L., Yu, M., Ramakrishna, S., Wang, Z.-L. & Long, Y.-Z. Insulator polymers achieve efficient catalysis under visible light due to contact electrification. *Water Res.* 226, 119242 (2022).
- [R3] Chen, B., Xia, Y., He, R., Sang, H., Zhang, W., Li, J., Chen, L., Wang, P., Guo, S., Yin, Y., Hu, L., Song, M., Liang, Y., Wang, Y., Jiang, G. & Zare, R. N. Water-solid contact electrification causes hydrogen peroxide production from hydroxyl radical recombination in sprayed microdroplets. *Proc. Natl. Acad. Sci. U. S. A.* 119, e2209056119 (2022).
- [R4] Dong, X., Wang, Z., Berbille, A., Zhao, X., Tang, W. & Wang, Z. L. Investigations on the contact-electro-catalysis under various ultrasonic conditions and using different electrification particles. *Nano Energy* 99, 107346 (2022).
- [R5] Li, F., Zhu, J., Sun, P., Zhang, M., Li, Z., Xu, D., Gong, X., Zou, X., Geim, A. K., Su, Y. & Cheng, H.-M. Highly efficient and selective extraction of gold by reduced graphene oxide. *Nat. Commun.* 13, 4472 (2022).

Comment 2. Basically, CEC belongs to mechanochemistry. Similar work related to mechanochemical reduction of metal ions have been reported (*J. Am. Chem. Soc.* 2015, 137, 5, 1726–1729, *Applied Materials Today* 26, 101339, *Mater. Horiz.*, 2016,3, 113-118).

Response: Thank you for your kind reminder. Although contact-electro-catalysis is a form of mechanochemistry, like ball milling, sonochemistry, piezocatalysis, tribocatalysis and magnetoelectric catalysis, the mechanism differs drastically from those.

Applied Materials Today 26, 101339, *Mater. Horiz.*, 2016, 3, 113-118 are both papers in which magnetoelectric material, magnetic micro/nano swimmers, are used as catalysis. The catalytic principle both these papers are based on is related to piezocatalysis, a phenomenon that involves the properties of materials in the bulk (rearrangement of charges in the lattice of the solid material causing the polarization of the surface) to convert a stress into catalytic reactions. Contact-electro-catalysis relies on the in-situ charging and discharging of the surface of the polymer particles involved in contact-separation cycles with a liquid, here water. It is a surface phenomenon (contact electrification) for which the ability of the material to withdraw charges by contact-electrification and emit electrons is conducive to the catalytic performances, as shown by the data presented in Figure 1 b,c,d the present manuscript (copied below for your convenience).

Figure R2. Copied from manuscript. (b) Reduction of AuCl₄⁻ to Au⁰ by CEC using various dielectric powders as catalysts. (c) Measurement of the charge of a microdroplet sliding on a slope made FEP, PTFE, PP or HDPE (d) Detection of TEMPO by EPR spectroscopy for various catalysts

On the other hand, in *J. Am. Chem. Soc.* 2015, 137, 5, 1726–1729, Baytekin et al. conducted an experiment based on research previously published by Liu and Bard (*Nature Mater* 7, 505–509 (2008)). These experiments rely on the much earlier discovered phenomenon that is tribocatalysis. As demonstrated previously by Jinyang Zhang et al. in *J. Am. Chem. Soc.* 2019, 141, 14, 5863–5870, in the case of electrochemistry on tribocharged polymers, i.e. solid-solid triboelectrification followed by immersion in a solution containing metal ions, the ability of a material to reduce metals is related not to the charge magnitude, contrary to the case of CEC, but to the ionization energy of the charged dielectric. The smaller the ionization energy the higher the catalytic activity. Moreover, in all of these papers treating the case of electrochemistry on tribocharged polymers, the metal is formed at the surface of the polymer. For the case of CEC, we obtained a precipitate of metal particles and did not observe any absorption of metals on FEP after the reaction, and we could recycle the material 5 times for a 20 hours long experiment.

To put it briefly, as mentioned in the response to the previous comment, contact-electrocatalysis is defined as a form of catalysis driven by the exchange of electrons produced by contact-electrification at solid-liquid interfaces (generally water and dielectric insulators). This concept should not be confused with tribocatalysis, which is the catalysis of reactions driven by charges generated on materials by solid-solid friction prior to contact with a liquid.

Contact-electro-catalysis is a form of mechanochemistry, it is true. However, we do not believe that the previous works are so similar to CEC reduction of metal ions in aqueous solutions that it should prevent the publication of the present work, or diminish its importance in the field of solid-liquid contact electrification and its applications.

Meanwhile, we also improved the introduction to help future readers understand how CEC should be distinguished from other forms of mechanochemistry, we made the following modification to the text, including a formal definition of CEC, and we added the following citations (see in red below, and highlighted in yellow in the main text):

Introduction:

“To this regard, public research in this area mostly focuses on methods based on solvent extraction⁶, selective precipitation⁷, or sieving by adsorption on various substrates⁸, or more rarely magnetoelectric materials.^{9,10}

Previously, various research groups explored the ability of dielectric materials, which have been pre-charged by solid-solid friction, to reduce metal ions in solution, i.e., the tribocatalytic reduction of metals^{11,12}. Later research showed that the catalytic ability of the dielectric insulator towards metal reduction is not correlated with its ability to withdraw charges; the catalytic activity of tribocharged materials is rather favored when the molecules that constitute the surface are characterized by a low ionization energy¹³. This phenomenon has been exploited by chemist to pattern surfaces with metal particles by rubbing two insulators together¹², or more recently to reduce CO₂ in presence of water by friction of copper on glass¹⁴, or trigger atomic transfer radical polymerization¹⁵.

Recently, we discovered that water-solid contact-electrification (SL-CE) can be used to catalyze chemical reactions¹⁶. This form of catalysis, called contact-electro-catalysis (CEC), takes advantage of electron exchanged by contact electrification during contact-separations cycles between a dielectric insulator and a liquid, generally an aqueous solution¹⁷, to catalyze redox reactions¹⁸. Since then, researchers found the effects of CEC can be obtained in various conditions including ultrasonic¹⁸ and microfluidic conditions¹⁹, or by sliding droplets on a slope under illuminations²⁰; CEC has also been proven to influence the outcome of some chemical reactions in microdroplets chemistry²¹. Moreover, it appeared that, contrary to tribocatalysis, the performances of dielectric insulators towards CEC at room temperature is correlated with their ability to withdraw electrons from water by contact-electrification.”

Results (Fig. 1)

“These observations suggest that the catalytic activity of the dielectric powders in CEC is linked to its ability to perform water-solid contact-electrification and to release those electrons. This confirms that the phenomenon at play here is indeed CEC, and not due to a solid/solid contact-electrification that is used to perform tribocatalytic reactions, as evocated in the introduction.¹³”

- [R6] Mushtaq, F., Chen, X., Veciana, A., Hoop, M., Nelson, B. J. & Pané, S. Magnetoelectric reduction of chromium(VI) to chromium(III). *Appl. Mater. Today* 26, 101339 (2022).
- [R7] Chen, X.-Z., Shamsudhin, N., Hoop, M., Pieters, R., Siringil, E., Selman Sakar, M., J. Nelson, B. & Pané, S. Magnetoelectric micromachines with wirelessly controlled navigation and functionality. *Mater. Horiz.* 3, 113–118 (2016).
- [R8] Baytekin, H. T., Baytekin, B., Huda, S., Yavuz, Z. & Grzybowski, B. A. Mechanochemical Activation and Patterning of an Adhesive Surface toward Nanoparticle Deposition. *J. Am. Chem. Soc.* 137, 1726–1729 (2015).
- [R9] Zhang, J., Rogers, F. J. M., Darwish, N., Gonçalves, V. R., Vogel, Y. B., Wang, F., Gooding, J. J., Peiris, M. Chandramalika. R., Jia, G., Veder, J.-P., Coote, M. L. & Ciampi, S. Electrochemistry on Tribocharged Polymers Is Governed by the Stability of

Surface Charges Rather than Charging Magnitude. *J. Am. Chem. Soc.* 141, 5863–5870 (2019).

Comment 3. If the authors can understand, I would like to ask them to consider the following points.

1. The description in the main text “CEC has been successfully employed for the degradation of organic dyes, the leaching of Li and Co from spent Li-Ion batteries cathode materials, and hydrogen peroxide evolution” is not comprehensive. Actually, CEC has been also employed in the fabrication of flammable gases and controlled radical polymerization.

Response: Thank you for the kind reminder. We would like to remind you of the definition of contact-electro-catalysis that has been previously proposed in our first publication in *Nature Comm.* and *Advanced Materials*, 2023, 202304387. Contact-electro-catalysis is the catalysis that is directly driven by contact-electrification at solid-liquid interface, the solid being a dielectric insulator.

When the reaction is driven by a solid-solid precharging of a dielectric insulator, the phenomenon is called tribocatalysis (See the work of Liu and Bard and others).

Concerning *Angew. Chem. Int. Ed.* 2023,62, 202309440, the paper reporting the tribochemically controlled ATRP, it is important to mention the difference in terms of both mechanism and process between their paper and CEC. Contact-electro-catalysis is defined as a form of catalysis driven by the exchange of electrons produced by contact-electrification at solid-liquid interfaces (generally water and dielectric insulators). This concept should not be confused with tribocatalysis, which is the catalysis of reactions driven by a pre-charging of a material by solid-solid friction prior to contact with a liquid (either while submerged or prior to submersion).

In the case of that paper, the authors are using a semiconducting material, TiO₂ to be specific, in friction with a polymer (stir bar) and the glass of the beaker, to involve reduce CuBr₂/Tris(2-Pyridylmethaylamine) which catalyze the ATRP reaction. In this case, the mechanism could be related to the tribocatalysis evocated above (see our description of CEC, as defined in our previous work published in *Nature Communications* in 2022, see below)

“Contact-electro-catalysis (CEC) was proposed as the mechanism for degrading MO in presence of FEP particles. The propagation of ultrasonic waves in solution provokes the formation of cavitation bubbles (CB). The collapse of cavitation bubbles is assumed to induce frequent contact electrification at the FEP-water interface, from which arises electron exchanges. The step-by-step illustrations are exhibited in Fig. 4a. A nucleus of CB is firstly formed during ultrasonication. Thereafter, the CB, containing dissolved gas, grows from the nucleus until reaching a critical size. At this point, the collapse of the CB creates a high-pressure microjet that chases the previously adsorbed water molecules on the FEP surface. An electron is transferred from water to FEP upon contact, and the notation of FEP is proposed to describe the charged state of FEP after separation from water. In the meantime, the enclosed O₂ is released and grabs the electron from the charged surface of FEP* once they collide. FEP* retrieves its initial uncharged state after exchanging this electron to O₂, and this cycle repeats itself as long as the emission of ultrasonic waves is sustained.”*

This said, although the description of the mechanism of CEC from the paper of 2022 involved contact-electrification at water-solid interface, it is clear that the definition we have proposed in our previous work might have been not clear or concise enough. Indeed, we discovered, through the reading of *Angew. Chem. Int. Ed.* 2023,62, 202309440, some authors might be confused by the concept of CEC and attribute it to other fields. For example, in the introduction of that paper the authors were citing a paper about piezocatalysis as reference 15 to refer to the generation of flammable gas by CEC, (*J. Phys. Chem. Lett.* 2010, 1, 6, 997–1002) but counted as CEC (see our previous response for more information).

Similarly, it is evocated in *Angew. Chem. Int. Ed.* 2023,62, 202309440 that CEC has been used for reducing metal ions in reference 14 (*J. Am. Chem. Soc.* 2023, 145, 5, 2800–2805), but the paper that is referred in that paper reports a microdroplet chemistry reaction. In that paper the authors state clearly that contact electrification is not involved in that process:

“It is worth mentioning that the interface between two different materials can lead to contact electrification. (63,64) To investigate this possibility, we replaced the silica capillary with a stainless-steel capillary which was grounded or biased between -5 V and $+5$ V, a voltage that far exceeds the reduction potential of the TMIs. Under all of these conditions, the reduced products were still observed. We conclude that the reduction is only a result of the microdroplet effect.”

They show the reduction of iron and copper ion in microdroplet is only due to the properties of microdroplets (gas-liquid interface driven spontaneous oxidation of hydroxyl that liberates free electrons) and not contact-electrification, which is reasonable considering we failed here to reduce them efficiently by CEC.

Meanwhile, concerning the generation of flammable gasses, we found a paper employing solid-solid tribocatalysis to produce flammable gasses, by Xiaodong Cui et al., that showed they could concomitantly convert H_2O and CO_2 by employing the friction between copper and glass underwater (another solid-solid case) (*ChemistrySelect*, 2023, 8, 202204146). It is still not contact-electro-catalysis, as it is not the sole friction of water on a solid that drives the reaction, but interesting nonetheless to mention in our introduction.

We are conscious that this kind of confusion may be stemming from a lack of clarity in the definition of CEC in our first publication in Nature Communications. To correct this problem and help future readers understand how CEC should be distinguished from other forms of mechanochemistry, and especially tribocatalysis, **we made the following modification to the text, which include a formal definition of CEC, and added the following citations (see in red below, and highlighted in yellow in the main text):**

Introduction:

“Previously, various research groups explored the ability of dielectric materials, which have been pre-charged by solid-solid friction, to reduce metal ions in solution, i.e., the tribocatalytic reduction of metals^{11,12}. Later research showed that the catalytic ability of the dielectric insulator towards metal reduction is not correlated with its ability to withdraw charges; the catalytic activity of tribocharged materials is rather favored when the molecules that constitute the surface are characterized by a low ionization energy¹³. This phenomenon has been exploited by chemist to pattern surfaces with metal particles by rubbing two insulators together¹², or more recently to reduce CO_2 in presence of water by friction of copper on glass¹⁴, or trigger atomic transfer radical polymerization¹⁵.

Recently, we discovered that water-solid contact-electrification (SL-CE) can be used to catalyze chemical reactions¹⁶. This form of catalysis, called contact-electro-catalysis (CEC), takes advantage of electron exchanged by contact electrification *during contact-separations cycles* between a dielectric insulator and *a liquid, generally an aqueous solution*¹⁷, to catalyze redox reactions¹⁸. Since then, researchers found the effects of CEC can be obtained in various conditions including ultrasonic¹⁸ and microfluidic conditions¹⁹, or by sliding droplets on a slope under illuminations²⁰; CEC has *also* been proven to influence the outcome of some chemical reactions in microdroplets chemistry²¹. *Moreover, it appeared that, contrary to tribocatalysis, the performances of dielectric insulators towards CEC at room temperature is correlated with their ability to withdraw electrons from water by contact-electrification. Since its discovery in 2022, ultrasonically assisted CEC has been successfully employed for the degradation of organic dyes¹⁶, using powders and films²² as catalysts, for the leaching of Li and Co from spent Li-Ion batteries cathode materials²³, and for hydrogen peroxide evolution¹⁸. All these processes were mostly oxidative processes, generally using polymers as catalysts at ambient temperature and in water, at the exception of the battery recycling experiment that was conducted at high temperature in presence of SiO₂, an acid, and at higher temperatures.*"

- [R1] Wang, Z., Berbille, A., Feng, Y., Li, S., Zhu, L., Tang, W. & Wang, Z. L. Contact-electro-catalysis for the degradation of organic pollutants using pristine dielectric powders. *Nat. Commun.* 13, 130 (2022).
- [R10] Berbille, A., Li, X.-F., Su, Y., Li, S., Zhao, X., Zhu, L. & Wang, Z. L. Mechanism for Generating H₂O₂ at Water-Solid Interface by Contact-Electrification. *Adv. Mater.* 35, 2304387 (2023).
- [R11] Wang, C., Zhao, R., Fan, W., Li, L., Feng, H., Li, Z., Yan, C., Shao, X., Matyjaszewski, K. & Wang, Z. Tribochemically Controlled Atom Transfer Radical Polymerization Enabled by Contact Electrification. *Angew. Chem. Int. Ed.* 62, e202309440 (2023).
- [R12] Hong, K.-S., Xu, H., Konishi, H. & Li, X. Direct Water Splitting Through Vibrating Piezoelectric Microfibers in Water. *J. Phys. Chem. Lett.* 1, 997–1002 (2010).
- [R13] Yuan, X., Zhang, D., Liang, C. & Zhang, X. Spontaneous Reduction of Transition Metal Ions by One Electron in Water Microdroplets and the Atmospheric Implications. *J. Am. Chem. Soc.* 145, 2800–2805 (2023).
- [R14] Cui, X., Wang, H., Lei, H., Jia, X., Jiang, Y., Fei, L., Jia, Y. & Chen, W. Surprising Tribo-catalytic Conversion of H₂O and CO₂ into Flammable Gases utilizing Frictions of Copper in Water. *ChemistrySelect* 8, e202204146 (2023).

Comment 4. Essentially CEC process is a heterogeneous catalytic process. The adsorption-desorption equilibrium is very important. What's the effect of ultrasound on this equilibrium?

Response: Thank you for this comment. It is true adsorption-desorption cycles are essential in many heterogeneous catalytic processes; however, the experiments we conducted (see below) didn't find it plays a key role in this reaction.

Considering it takes 20 - 60 min to reach an equilibrium in non-disturbed aqueous solution, we decided to conduct an intermittent experiment by turning off the ultrasound after 30 min of ultrasonication, and turning it back on after 30 min of pause. Samples were taken at 0 min, after

stopping the ultrasounds and before starting the ultrasounds. The result is then compared to a continuous experiment.

It is well known that cavitation can increase the adsorption process (*Ultrasonics Sonochemistry, Volume 75, July 2021, 105610*). A slight increase in $[\text{AuCl}_4]^-$ concentration (decrease of the gold extraction value) was found after 30 min of OFF state; this could come from the desorption of a small amount of $[\text{AuCl}_4]^-$ from FEP powder. Meanwhile it is worth mentioning that this change is below the ppb level.

We conducted this experiment for 2 different power values of the ultrasonic bath and made a similar observation.

Figure R3. Intermittent and continuous reaction in 120 W ultrasonication.

Table R1. Intermittent and continuous result for 120 W 40 kHz and 60 W 40kHz ultrasonication.

Time (min)	Gold extraction rate (%)	
	40 kHz, 120 W	40 kHz, 60 W
30	44.37197	19.68413
60	43.88840	19.13771

90	77.65794	32.96148
120	77.00936	32.39498
60 (continuous)	77.75359	33.03139

This observation could be explained by the low specific surface area of the FEP powder used here. The isotherm obtained by the BET method shows a typical type II behavior (see below). The pore size distribution, calculated based on the HK method, shows an average pore size width of ~1.3 nm. Finally, multipoint BET surface area and total pore volume for pores are of only 9.023 m²/g and 0.0288 cm³/g.

Figure R4. BET isotherm plots for nitrogen adsorption capacity of 2 μm FEP. The insert is pore distribution

To take into account your remarks, the above figures and tables have been added to the manuscript:

Results:

*“Finally, to exclude the possibility that the decrease of gold concentration could result from a potential ultrasonic absorption of gold on FEP, we conducted an intermittent experiment (30min US ON, 30min US OFF) to evaluate the influence of adsorption/desorption cycles. Whether a power of 120 W or 60W is used, we only observe a slight desorption of gold, below ppb level, after the adsorption equilibrium is reached (30min), between each ultrasonication cycles, see Fig. S2 and table S1. The result of the intermittent experiment compared with that of the continuous experiment, shown in **Supplementary Fig. 2**, do not show a significant difference between the two setups. These results could be explained by the low surface area (9.023m².g⁻¹) and total pore volume (0.0288 cm³.g⁻¹) of the particles employed (see*

Supplementary Fig. 3, and Supplementary Table 1 and 2). These observations confirm that CEC is the main driving mechanism behind the extraction of gold.”

-[R15] Naidji, B., Hallez, L., Et Taouil, A., Rebetez, M. & Hihn, J.-Y. Effect of cavitation intensity control on self-assembling of alkanethiols on gold in room temperature ionic liquids. *Ultrason. Sonochem.* 75, 105610 (2021).

Reviewer #2 (Remarks to the Author):

This manuscript reported a metal free mechano-catalytic methods to efficiently degrade refractory organic compounds. The authors also designed a metal-free, selective, and recyclable catalytic gold extraction methods from e-waste aqueous leachates. This work is interesting. Still, there are some issues need to be addressed before the consideration of publication:

We thank reviewer #2 for taking the time to evaluate the present manuscript, and their comment and remarks. We are confident we addressed the concerns of Reviewer #2 appropriately and improved the manuscript based on their comments.

Comment 1. The authors discussed the recyclability of the FEP catalyst and drew the conclusion for only one work cycle. More work cycles need to be conducted and also long-term stability of this catalyst need to be tested.

Response: Thank you for your kind reminder. However, we would like to bring to your attention that we already answered both of these questions.

In the legend figure 4, that you can find below for your convenience, we clearly show that the same particles were reemployed for 5 works cycle, each cycle lasting 18 hours, which corresponds to a total of 90 h; this experiment was reproduced 3 times (error bars are present on the figure).

The fact the gold extraction rate barely changing between each cycle, in the very harsh conditions in which this experiment was conducted (low pH), illustrates very well the catalyst is stable. This is quite expected for a polymer that is chemically inert (FEP) and is consistent with previous experiments.

We added the following sentence to Figures 1 and 2 to define the error bars:

Figure 1 and 4:

“Fig. 1. Reduction of metal ions in solution by ultrasonication in presence of fluorinated ethylene propylene (FEP) microparticles. (a) Graphical description of the experimental setup and observations. (b) Reduction of $AuCl_4^-$ to Au^0 by CEC using various dielectric powders as catalysts. (c) Measurement of the charge of a microdroplet sliding on a slope made FEP, PTFE, PP or HDPE (d) Detection of TEMPO by EPR spectroscopy for various catalysts (e) Schematic description of the reduction of various metal ions (Mn^+) in aqueous solution by ultrasonically driven CEC in presence of FEP. (f) Gold extraction capacity for FEP-based CEC compared to adsorption/reduction method using different materials^{8,18–31}, on a range from around 0.1 ppm to around 100 ppm. (g) Reduction rate of various metal ions in aqueous solution by FEP-based CEC after 1 h. Error bars represent standard deviations for 3 reproduced experiments.”

“Figure 4. Selective extraction of Gold from the leachate of spent CPUs by CEC. (a) Schematic of the pre-treatment of CPU (blue) and recovery of CEC for extracting gold from spent CPUs. (b) Evolution of gold extraction over time from a leachate of CPU at a concentration of 10 ppm of gold. (c) Amount of $AuCl_4^-$, Zn^{2+} , Fe^{2+} , Ni^{2+} , and Cu^{2+} reduced after 20 h. (d) Recycling of FEP particle for 5 cycles of 18 h each. Error bars represent standard deviations for 3 reproduced experiments.”

Comment 2. The energy conversion efficiency of this CEC catalysis should be calculated and compared to the traditional methods.

Response: Thank you for this comment. The overall energy conversion comprises the mechanical conversion efficiency ($\eta_{\text{ele-sono}}$) and sonochemical efficiency ($\eta_{\text{sono-chem}}$). For the former, calorimetric measurements are most commonly used for quantifying the dissipated ultrasonic power. The energy balance is expressed as:

$$P_{\text{cal}} = P_{\text{out}} + P_{\text{accum}}$$

where P_{cal} is the ultrasonic energy dissipated into water determined calorimetrically and P_{accum} is the thermal energy accumulated in the body of water during ultrasonication. The accumulated energy in water is calculated as:

$$P_{\text{accum}} = mC_p dT/dt$$

where m is the mass of the irradiated water (2 kg), C_p is the heat capacity per unit mass of water (J/kg K) and dT/dt is the slope of the temperature rise versus time (K/s).

Precise measurement requires a customized closed cavitation chamber and a large number of sensors (more than 40, see *Ultrasonics*, 2015, 57, 18-30). Here we assume $P_{\text{out}}=0$ and take dT/dt as the average increased temperature (22 K) in 30 minutes to simplify the measurement. We found $P_{\text{accum}}= 102.67\text{W}$. So:

$$\eta_{\text{ele-sono}} = P_{\text{accum}}/P_{\text{total}} = 85.6\%$$

$$\eta_{\text{sono-chem}} = (r_{\text{metal}} \times \Delta H_{\text{metal}})/P_{\text{accum}}$$

Where r_{metal} is metal reduction rate and ΔH_{metal} is the reaction enthalpy of the reduction of M^{n+} to M , C_0 is the initial concentration of M^{n+} , C is the concentration of M^{n+} at moment t , t is the reaction time.

We take Au for example, with $\Delta H_{\text{AuCl}_4^-} = -322.168 \text{ kJ/mol}$ (at 298.15 K), $r_{\text{metal}}=0.08 \text{ mmol/h}$ (at 0.1 mM KAuCl_4).

We then obtain $\eta_{\text{sono-chem}}= 0.00697\%$.

It is worth noting this efficiency is highly dependent on solution's gold concentration. For example, if we change the concentration to 1 mM KAuCl_4 , then $\eta_{\text{sono-chem}}=0.0349\%$.

We would like to express that comparing CEC with traditional methods through this energy conversion efficiency might not be suitable, since these methods are using reductive reagents rather than catalyst; these processes are not converting mechanical energy into chemical energy.

Similarly, comparison with gold adsorbents would be meaningless as those, similarly, do not use a source of energy during the reduction process, or the adsorption process. For example, for nanoporous materials, such as metal-organic framework, covalent organic polymer, and active carbon. The gold extraction behavior of these materials is mainly attributed to the immobilization of gold ions within the pores, and the chemical reduction of the gold ion by the added functional groups, if applicable. For this kind of material, we cannot calculate its energy conversion efficiency because there is no clear energy conversion path. Moreover, these processes require a further desorption and/or leaching of the gold from the sorbent, followed by a reprecipitation; those steps are generally not considered in these papers. Therefore, comparisons would be difficult with CEC, a process that does not require those steps.

The other set of gold adsorbents, for example, 2D MoS₂, amyloid, cyclodextrin, their efficient gold extraction is accounted to the chemical reduction of gold ions to Au⁰ by photoreduction or with the functional groups in the pores; gold is embedded within the adsorbents. Among this kind of materials, only a few studies are suitable for using energy conversion efficiency as a point of comparison. We could select those with photoreduction ability to compare with CEC. The table is shown below.

Table R2. Literature report about comparing energy conversion of CEC and traditional methods.

Materials	Strategy	Types of gold ion	Gold ion concentration (ppm)	Energy conversion efficiency (%)	References
MoS ₂ /ZnS nano-heterojunction	Adsorption + photoreduction	[Au(S ₂ O ₃) ₂] ₃ (ΔH = -514.9 kJ/mol)	100	0.00090381	10.1016/j.cej.2020.124866
ZnS				0.00019993	
MoS ₂				0.00052546	
Mn ₂ O-MoS ₂	Adsorption + photoreduction	[Au(S ₂ O ₃) ₂] ₃	100	0.00113981	10.1016/j.jallcom.2022.167185
O-MoS ₂				0.00090823	
Mn-MoS ₂				0.00068629	
Graphite carbon nitride	Adsorption + photoreduction	AuCl ₄ ⁻	78.788	0.00034735	10.1039/C4TA04400B
FEP	CEC	AuCl ₄ ⁻	19.697	0.00596632	This work

In this case, although the CEC process seems quite favorable if we consider that photocatalytic processes would be employed indoor and using a lamp as a light source, we still consider it to be off-topic since photocatalysis are supposed to be used to convert sunlight energy, ultimately.

-[R15] Naidji, B., Hallez, L., Et Taouil, A., Rebetez, M. & Hihn, J.-Y. Effect of cavitation intensity control on self-assembling of alkanethiols on gold in room temperature ionic liquids. *Ultrason. Sonochem.* 75, 105610 (2021).

- [R16] Zhan, W., Yuan, Y., Yang, B., Jia, F. & Song, S. Construction of MoS₂ nano-heterojunction via ZnS doping for enhancing in-situ photocatalytic reduction of gold thiosulfate complex. *Chem. Eng. J.* 394, 124866 (2020).
- [R17] Liang, Y., Zhan, W., Yuan, Y., Zamora-Romero, N., Jia, F., Yang, B., Nasrddinov, Z., Arauz-Lara, J. L. & Song, S. Manganese and oxygen dual-doping MoS₂ boosts reduction and adsorption activity toward efficient recovery of gold(I) from thiosulfate solutions. *J. Alloys Compd.* 928, 167185 (2022).
- [R18] Guo, Y., Zhang, L., Zhou, K., Shen, Y., Zhang, Q. & Gu, C. Selective gold recovery by carbon nitride through photoreduction. *J. Mater. Chem. A* 2, 19594–19597 (2014).

Comment 3. The catalysts used in this work, like FEP, PTFE, PP, and HDPE, are all hydrophobic. When they were soaked in aqueous solution, is there any chance for them to bring air in the reaction? Will the air be involved in the reduction?

Response: Thank you for your kind reminder. For your first question, we carefully checked the oxygen in the solution with a dissolved oxygen meter (Smart Sensor type AR8210 with optical probe). We found that in our 50 ml solution, the dissolved oxygen concentration decreased to less than 0.01 mg/L after 90 seconds of N₂ bubbling (~100 ml/min). In addition, the BET results (see comment 5) also showed that the FEP employed is endowed with a small surface area and porosity.

The isotherm obtained by the BET method shows a typical type II behavior (see below). The pore size distribution, calculated based on the HK method, shows an average pore size width of ~1.3 nm. Finally, multipoint BET surface area and total pore volume for pores are of only 9.023 m²/g and 0.0288 cm³/g.

Figure R4. BET isotherm plots for nitrogen adsorption capacity of 2 μm FEP. The insert is pore distribution

Based on those we could assume that the dissolved oxygen concentration is negligible after 15 min bubbling. Meanwhile, as seen in Figure 2g, the reduction of gold and other metals is still occurring in aerobic conditions, albeit at a lower rate; indeed, in presence of oxygen, a competition between the oxygen reduction reaction and the metal reduction reaction is taking place. In presence of oxygen ROS and H₂O₂ are generated, as reported previously in *Nat Commun* 13, 130 (2022), and *Adv. Mater.*2023,35, 2304387.

-[R1] Wang, Z., Berbille, A., Feng, Y., Li, S., Zhu, L., Tang, W. & Wang, Z. L. Contact-electro-catalysis for the degradation of organic pollutants using pristine dielectric powders. *Nat. Commun.* 13, 130 (2022).

-[R10] Berbille, A., Li, X.-F., Su, Y., Li, S., Zhao, X., Zhu, L. & Wang, Z. L. Mechanism for Generating H₂O₂ at Water-Solid Interface by Contact-Electrification. *Adv. Mater.* 35, 2304387 (2023).

Comment 4. This work only presented the catalysis in aqueous solution. How about other solvent?

Response: Thank you for this comment. As far as we know, no existing literature reported a successful experiment conducting contact-electro-catalysis in organic solvent. All the former studies were based on aqueous solutions. Here we tested DMSO, ethanol and acetonitrile as solvent to reduce metal. DMSO, an aprotic solvent, was chosen for its high polarity, and ability to dissolve both polar a non-polar compound. Ethanol was chosen as a polar protic solvent, with a lower polarity than water but still possessing the ability to form hydrogen bounds. Acetonitrile, a polar aprotic solvent, has been selected mainly because a former study (*Nature Sustainability* 4, 618–626 ,2021) employed it to leach precious metals from e-waste, which makes it a potential medium from which we could have extracted gold ions and reduce them to solid particle by CEC. However, in stark contrast with the experiments using DI water as a solvent, we observed no change in gold concentration outside the ICP-OES error range in these three solvents after 1 hour of ultrasonication in N₂ atmosphere in presence of FEP as the catalyst. The data is presented in the figure below.

Figure R5. AuCl₄⁻ reduction by CEC in DMSO, acetonitrile, ethanol, and water.

This observation could stem from the weaker solid-liquid contact-electrification ability of organic solvent compared to that of water; water possesses a very developed hydrogen bond network that allows for a fast reconfiguration of charges, which could in turn be one of the origins of its superior contact-electrification abilities (*ACS Nano* 2021, 15, 9, 14830–14837, *Adv. Mater.*2023,35, 2304387.)

We added the following comment in the manuscript to take into account this observation and the above figure was added to the supplementary information as Supplementary Figure 6:

*“The experiment was also conducted in other solvents than water, namely dimethyl sulfoxide (DMSO), Acetonitrile, or ethanol, and in the same conditions of temperature, power, concentration and catalyst loading as for water. For all these 3 solvents, we did not observe any significant decrease of the concentration of the AuCl₄⁻ (see **Supplementary Fig. 6**).”*

- [R19] Chen, Y., Xu, M., Wen, J., Wan, Y., Zhao, Q., Cao, X., Ding, Y., Wang, Z. L., Li, H. & Bian, Z. Selective recovery of precious metals through photocatalysis. *Nat. Sustain.* 4, 618–626 (2021).
- [R20] Zhang, J., Lin, S., Zheng, M. & Wang, Z. L. Triboelectric Nanogenerator as a Probe for Measuring the Charge Transfer between Liquid and Solid Surfaces. *ACS Nano* 15, 14830–14837 (2021).
- [R10] Berbille, A., Li, X.-F., Su, Y., Li, S., Zhao, X., Zhu, L. & Wang, Z. L. Mechanism for Generating H₂O₂ at Water-Solid Interface by Contact-Electrification. *Adv. Mater.* 35, 2304387 (2023).

Comment 5. As we know, the triboelectric induced charge density is highly related to the surface property between two friction layers more than the intrinsic property. In this case, please discuss the effects of the factors, such as the particle size and the porosity of FEP, to the catalysis process.

Response: Thank you for this comment. Based on the literature and experimental data, we found that the chemical property of dielectric powder is more important than physical property in CEC reaction.

Here, we used 10 mg of FEP powder with a diameter of 0.2, 2, 6.5, 15, 30 μm to reduce 0.1 μM [AuCl₄⁻] in anaerobic environment. The result as well as BET isotherms, size distribution characterization of each powder are presented below.

Table R3. Surface properties of different sizes of FEP powder.

Size (um)	Specific surface area (m ² /g)	Total pore volume (cm ³ /g)	Average pore diameter (nm)	Porosity (ρ=2.15 cm ³ /g)
0.2	88.458	0.24970	11.290	0.536877
2	9.023	0.02880	12.770	0.061920
6.5	5.159	0.02356	18.260	0.050654
15	4.017	0.01005	10.000	0.021608
30	3.997	0.00990	9.906	0.021285

Figure R5. The size distribution of 0.2, 2, 6.5, 15, and 30 μm FEP powder.

We found surface area does help increase reduction efficiency. The increase is more obvious in the range of 30 um-2 um. When it goes to nanometer level, the rise in efficiency is not significant.

Figure R6. The gold extraction rate of 0.2, 2, 6.5, 15, and 30 µm FEP powder.

Porosity is always related to surface area. Due to FEP's chemical inertness, it is hard to use acid to etch its surface. We previously employed ICP etching, using successively Ar and SF₆ gasses, to test the effect of surfaced area on a film's ability to perform CEC, see Nano Research, 2023, s12274-023-6125-9. In this experiment, we found that increasing the surface area of the film catalyst had very little influence compared to the change in the chemical composition of the surface. This result confirmed the hypothesis formulated in Nanoscale 15, 6243–6251 (2023), which was based on the XPS results, according to which the increased activity of the etched FEP film was more likely due to the increase in relative concentration of -CF₃ groups at the tip of the surface structures generated by Ar plasma etching, rather than the increase of surface area.

[R21] Su, Y., Berbille, A., Wang, Z. L. & Tang, W. Water-solid contact electrification and catalysis adjusted by surface functional groups. *Nano Res.* (2023) doi:10.1007/s12274-023-6125-9.

[R22] Zhao, X., Su, Y., Berbille, A., Wang, Z. L. & Tang, W. Degradation of methyl orange by dielectric films based on contact-electro-catalysis. *Nanoscale* 15, 6243–6251 (2023).

We added the following comments to consider this point:

Results:

“In Supplementary Fig 5a and 5b, particles of size were tested, showing that the catalytic efficiency improves as the particles size decreases from 30µm to 6.5µm; however, it only marginally increased from 6.5 µm to 0.2µm. Interestingly, despite a large increase of the specific surface area and porosity from particles of 2 µm to 0.2 µm, see Supplementary Table 2, we did not observe a proportional increase of the gold extraction efficiency. This could be explained by the fact that, as demonstrated by a previous experiment, the surface composition is more conducive to the CEC performance of a material than its surface area.²⁴”

-[R23] Su, Y., Berbille, A., Wang, Z. L. & Tang, W. Water–solid contact electrification and catalysis adjusted by surface functional groups. *Nano Res.* (2023). doi:10.1007/s12274-023-6125-9

Comment 6. If the reduction happened in multi-ions high conductive solution, like 0.1~1 M NaCl solution, will the CEC catalysis still be effective?

Response: Thank you for your question. The question of whether or not this process can be carried out in solution with high ionic strengths has been addressed in Supplementary Figure 2c, mentioned in this passage of the manuscript that we included below. It seems the way we presented this part of the manuscript was not explicit enough. To ensure future readers will not miss this detail, we added the reference to the specific figure when mentioning the various parameters in the text (modifications are in red in the text below and highlighted in yellow in the manuscript).

Results:

“Prior to further experiments with various metal ions, we optimized the conditions of the reaction by successively studying the effects of catalyst loading, temperature, ionic strength, pH and *particle size* on the reduction of AuCl_4^- by CEC, for 1h (**Supplementary Fig. 4 and 5**). The observations are mostly consistent with previous literature reporting the performances of FEP for CEC^{18,38}. The optimal catalyst loading is 1:5000 (FEP:Water) (**Supplementary Fig. 4a**) and the temperature is kept at 25 °C (**Supplementary Fig. 4b**). Because of the charge screening effect arising under high ionic strength conditions¹⁷, which results in a decreased electron exchange by water-solid CE, adding NaCl to the solutions slows down the reaction (**Supplementary Fig. 4c**). Similarly, adjusting the pH of the AuCl_4^- solution results in a decrease of the catalytic activity owing to the consequential increase of ion concentration (**Supplementary Fig. 4d**).”

For your convenience, we copied Fig. S2c below.

Figure R7. The influence of NaCl concentration on the reduction of the AuCl₄⁻.

The figure S2c shows that the reaction is still occurring as salt concentration increases, although at a much slower rate when the concentration of sodium chloride exceeds 100 mmol L^{-1} . At 1 mol L^{-1} , however, the charge screening effect mentioned in the text becomes so strong that electron transfers during contact-electrification is not effective. As a consequence, above 1 mol L^{-1} , the reaction could hardly be called effective. However, although we conducted these experiments in order to understand better our reaction system, it is worth mentioning that a concentration of sodium chloride at 1 mol L^{-1} is not something one would meet in a practical setup. To see an experience that is closer to cases of e-waste recycling, we would like to encourage you to read the paragraph below that explains a little more about the Fig. 4 of our main text.

Concerning the ability of CEC to perform metal reduction in multi-ions solution, this point is addressed as well in Figure 4b and c in the main text (also reported below). Here, the experiment is conducted in a solution with a low pH (due to the acid leaching method employed to dilute metals from CPU or Electroplating waste material). In both cases, the system contains multiple metal ions. This experiment shows that not only CEC can reduce gold in a multi-ion system, but that it is a selective process (details about the selectivity mechanism are reported in the response to question 7).

Figure 4. Selective extraction of Gold from the leachate of spent CPUs by CEC. (a) Schematic of the pre-treatment of CPU (blue) and recovery of CEC for extracting gold from spent CPUs. **(b)** Evolution of gold extraction over time from a leachate of CPU at a concentration of 10 ppm of gold. **(c)** Amount of AuCl_4^- , Zn^{2+} , Fe^{2+} , Ni^{2+} , and Cu^{2+} reduced after 20 hours. **(d)** Recycling of FEP particle for 5 cycles of 18h.

Comment 7. Please explain the working mechanism of the selective reduction of Au in the CPU leachates.

Response: Thank you for raising this question, thus giving us the opportunity to develop this point. The selectivity mechanism for the reduction of gold in a leachate obtained from CPU by acid leaching is attributed to a large difference in kinetics between the reduction reaction of gold and those of the other metals.

In physical chemistry, it is generally considered there are two main types of competing mechanism to lead to selectivity of various chemical reactions. One of them is thermodynamic, and the other is a kinetic (*Angew.Chem.Int.Ed.*2006,45,4724–4729).

- a) If the competing reactions are mostly irreversible, and if their rate is only time dependent, then the control of the reaction is kinetic.
- b) If the competing reactions are reversible, but products can revert to the starting materials through another path, and/or if the products can convert through another independent process, then we have a thermodynamically influenced or thermodynamically controlled competition.

In our present experiment, considering the solution is neutralized after the leaching to prevent redissolution of solid precipitate (see method), we can assume that the selectivity is mostly resulting from the kinetic of the reaction. To verify this hypothesis, we conducted the reduction of Zn, Fe, Cu, Ni in DI water, similarly to the experiments presented in Figure 1g. Below you will find the figure comparing the % of metal reduced for each metal ions dissolved in DI water, at 25 °C , and exposed to 120 W, 40 kHz ultrasound for 1 hour in separate experiments.

Figure R8. Fe, Ni, Cu, Zn, and Au extraction rate in their anaerobic aqueous solution.

We included an explanation of this mechanism in the main text (see below change reported in red, and highlighted in yellow in the text).

Results:

“This selectivity could result from a kinetic mechanism of selectivity, owing to the fact the solution was neutralized to avoid redissolution of the metals, i.e., reversible reactions⁴¹. To ensure it was the case, prepared individual solutions of each of the metal ions present in the leachate as individual solutions

*in DI water; the solutions are then ultrasonicated (40 kHz, 120 W) in presence of 10 mg of FEP, at 25 °C. We observed that, even when they are reduced in separate solution, while a great portion of the gold ions are reduced, Fe, Ni, Cu, Zn ions are barely removed from the solution after 1 hour of treatment (see **Supplementary Fig. 18**)."*

-[R24] Berson, J. A. Kinetics, Thermodynamics, and the Problem of Selectivity: The Maturation of an Idea. *Angew. Chem. Int. Ed.* **45**, 4724–4729 (2006).

REVIEWER COMMENTS

Reviewer #1 (Remarks to the Author):

The authors have addressed most of my concern. This revised version has presented a more straightforward description of the novelty for this work. The manuscript is recommended to be accepted after the following specific questions can be addressed.

1. PP reveals a similar structure with PE, why PP shows much stronger CEC reduction ability?
2. What's the effect of macromolecular architecture on CEC reduction ability, such as isotactic PP vs atactic PP et al.
3. Additives (antioxidants, plasticizer, inorganic fillers et al.) are normally presented within most industrial polymer products. There was no purification process prior to the CEC experiment as described in the manuscript. Will these additives possibly affect the CEC catalytic efficiency since CEC is highly sensitive to the ionic concentration in water?
4. More information for these commercial polymers (such as specification, MW...) should be provided.
5. Why Pd was chosen as the model ion for calculation rather Au?
6. It's not easy for maintaining N₂ atmosphere constantly in beaker under ultrasound. More details are required for the reaction setup.

Reviewer #2 (Remarks to the Author):

All the comments have been well addressed to improve the manuscript. The revised version was suggested to be accepted.

We warmly thank the reviewers for their overall positive outlook on the present version of the manuscript. Reviewer #1 raised supplementary questions concerning some aspects of the work. We addressed their questions, conducted supplemented experiment, and modified the manuscript accordingly. The quality of the manuscript has been greatly improved, and we thank both reviewers for their dedication.

Reviewer #1 (Remarks to the Author):

The authors have addressed most of my concern. This revised version has presented a more straightforward description of the novelty for this work. The manuscript is recommended to be accepted after the following specific questions can be addressed.

We thank you for your evaluation of the revised manuscript. We considered the points you suggested following the first revision and addressed them either experimentally or by computation when we saw fit. Revisions brought to the manuscript are reported below in “red”, and highlighted in yellow in the marked copy of the revised manuscript.

1. PP reveals a similar structure with PE, why PP shows much stronger CEC reduction ability?

Thank you for the question. In the Figure 1 b, c and d, we reported the ability of different polymers towards, respectively, performing the reduction of gold from a synthetic solution of AuCl_4^- , extracting negative charges from water by solid-liquid contact electrification, and subsequently releasing electrons in the solution.

In figure 1b and 1c, we see that HDPE has a lower ability to extract negative charges from water than all polymers, including PP. In theory, if the negative charges are extracted in the form of electrons, it would mean that less electrons would be released in the solution during an experiment of contact-electro-catalysis. In Figure 1c, we observe that, indeed, HDPE as a contact-electro-catalyst is releasing significantly less electrons than other materials, and almost as much as the control (no catalyst). All these observations show a correlation between the ability of polymers to extract and release electrons by CE at water-solid interface, and their ability to perform a reduction reaction in aqueous solution.

Figure R1. The structure of PE (polyethylene) and PP (polypropylene).

The difference between PP and HDPE is that while HDPE comprises only hydrogen functional groups, PP comprises methyl groups, as shown in Figure R1. In previous research, we have shown that methyl groups are able to extract negative charges during contact electrification more efficiently than simple

hydrogen atoms (“The methyl group has a greater ED [(electron withdrawing)] ability than the hydrogen group” in Shuyao Li, Jinhui Nie, et al., *Adv. Mater.* 2020, 2001307).

[R1] Li S, Nie J, Shi Y, Tao X, Wang F, Tian J, Lin S, Chen X & Wang ZL. Contributions of Different Functional Groups to Contact Electrification of Polymers. *Adv Mater.* 2020 Jun;32(25):e2001307.

2. What’s the effect of macromolecular architecture on CEC reduction ability, such as isotactic PP vs atactic PP et al.

We thank you for suggesting to examine this aspect; this is an interesting question. From the understanding we gathered over the years concerning electron transfers by contact-electrification, in the case of PP, we should not see a massive difference in the electron-withdrawing ability from water between isotactic, atactic and syndiotactic PP. This is mostly assumed as such because our previous experiments showed that this property is mainly derived from the type of functional group present at the surface of the solids. Meanwhile, we cannot simply rely on assumptions; therefore, we employed computation to the ability of atactic PP, isotactic PP and syndiotactic PP to perform oxygen reduction reaction.

Figure R2. The structure of isotactic, syndiotactic and atactic PP.

The computational results presented below show that for a fixed bond length the adsorption energy between an oxygen molecule and isotactic, syndiotactic or atactic polypropylene are similar; this result is conducive to a negligible difference in the amount of energy required for the electron transfer from PP to oxygen (oxygen reduction reaction).

Figure R3. Simulated energy difference $\Delta E(\text{HOMO}_{(\text{PTFE})} - \pi(\text{O}_2))$ of isotactic, syndiotactic and atactic PP. Carbon: purple, Hydrogen: blue, Oxygen: red.

Although the result is a negative (we see barely any difference between the different tacticity), we decided to include this result in the manuscript as followed :

“What’s more, using polypropylene as a model we found that tacticity does not have a significant influence on the reduction process by CEC; at a fixed bond length of 2.11 Å, $\Delta E(\text{HOMO}(\text{PTFE}) - \pi(\text{O}_2))$ of isotactic, syndiotactic and atactic PP reach, respectively, 1.83 eV, 1.83 eV, and 1.82 eV (see Supplementary Figure 19).”

3. Additives (antioxidants, plasticizer, inorganic fillers et al.) are normally presented within most industrial polymer products. There was no purification process prior to the CEC experiment as described in the manuscript. Will these additives possibly affect the CEC catalytic efficiency since CEC is highly sensitive to the ionic concentration in water?

Thank you for bringing up this point to our attention. We took additives into account from the beginning of our experiment; hence we have chosen pure resins as our main studied polymers. FEP is Teflon FEP 100 from Dupont². Their datasheets states that “Teflon FEP 100 is a melt-processible copolymer of tetrafluoroethylene and hexafluoropropylene, without additives that meets the requirements of ASTM D 2116 Type I.”. Our PTFE is Teflon PTFE 7A X from Dupont³ that meets the requirements of ASTM D4894-15, Type II (“The materials included herein do not include mixtures of PTFE resin with additives such as colorants, fillers or plasticizers; nor do they include reprocessed or reground resin or any fabricated articles”). PP and HDPE are from Macklin and Sigma-Alrich, respectively, but we didn’t find information about additives content. However, we conducted an experiment to demonstrate the influence of some commonly encountered additives on CEC.

As far as we know, these additives are very unlikely to leak in the water in such magnitude that it would significantly affect the reactions, at least we never observed such effects, even in our recycling experiment in the present paper, for which the particles were washed between experiment. Meanwhile, we understand that such case should be considered, since even sodium chloride diluted in water to concentrations in the nanomolar range seems to affect the experiment. Therefore, we conducted supplementary experiments to address this point.

We chose plasticizers, antioxidants and reinforcing agents to check their effects on CEC gold reduction when there are present in trace amounts (100 ppb), which would correspond to an important leak of

additives. 2,6-Di-tert-butyl-4-methylphenol (BHT) is a common antioxidant which can be used as a representative of phenolic antioxidants. Phthalates are often used as plasticizers, for example, di(2-ethylhexyl)phthalate (DIP). Inorganic fillers such as SiO₂ mica has always been employed to increase intensity.

It appears that in presence of these additives in the water, the efficiency is barely changed (see figure below).

Figure R4. Reduction of 0.1 mM AuCl₄⁻ in 100 ppb various additives including SiO₂, BHT, and DIP, in the presence of 10 mg FEP after one hour. The control is a standard experiment in absence of any additive in the solution.

We added the following modifications to the manuscript to take into account your remarks and added the data to the SI as Supplementary Fig. 7.

Main Text:

*“The FEP resin our particles are made from does not contain additives (see Method section); however, we considered the possibility that leaking from other commercial polymers could influence the reaction, considering the sensitivity of CEC to ionic species and contaminations. In **Supplementary Fig. 7**, the results of gold reduction experiments in presence of trace amount of additives (~10² ppb) in the solution show that additives do not affect the CEC gold reduction process .”*

The method was modified to give the exact reference of the FEP and PTFE we purchased.

Method:

*“Fluorinated ethylene propylene (FEP 100, Dupont, Mn = 676 000 000), Polytetrafluoroethylene (PTFE 7AX, Dupont, Mn = 316 000 000), Polypropylene (PP, Macklin, Mw = 289 300, (see **Supplementary Fig. 22**), High-density polyethylene (HDPE, Sigma-Aldrich, Mw = 3 000 0000 ~ to 6 000 000),...”*

[R2] <https://www.teflon.com/en/-/media/files/teflon/teflon-fep-100-product-info.pdf?rev=c37d94036ad04b33a7a44325ac07a3c1&hash=15E3B14B9236C1F52CE5FE80CFF6C8C4>

[R3] <https://www.teflon.com/en/-/media/files/teflon/teflon-ptfe-7a-x-product-info.pdf?rev=8857d802e22f4b08b7b68b7ef9da7b3d&hash=B453CCA54062515E454FD2598269C554>

4. More information for these commercial polymers (such as specification, MW...) should be provided.

Thanks for this comment. We understand with your demand of more information concerning the polymer characterization. In the method sections we previously reported the material and manufacturer of the different polymers used here, which gives the readers the opportunity to look for the datasheets of the company; however, we omitted to precise which references were purchased. This issues as been fixed. We also employed Gel Permeation Chromatography, combined thermogravimetric and differential scanning calorimetry analysis (TG-DSC) and Fourier transformed infra-red spectroscopy (FTIR) to gather more data or complete those we could not obtain from the manufacturers.

GPC analysis is not applicable for FEP and PTFE, due to their superior chemical stability. Standard specific gravity (SSG) and melt flow rate (MFR) are more conveniently and frequently used, rather than rheological and dynamic scattering methods. The data of these two experiments are extracted from the datasheets of Dupont. Molecular weight can be calculated from SSG or MFR. Take SSG as sample⁴:

$$\overline{LogMn} = \frac{2.61 - SSG}{0.06}$$

The data was reported in Supplementary Figure 22- 24. We put the key information as following:

Tabel R1. Information about FEP, HDPE, PP and HDPE.

Name	M	D	Molecule Weight (10 ⁵)
FEP	264.18	588.26	Mn=676
HDPE	176.20	457.59	Mw=30-60
PP	169.23	456.49	Mw=2.893
PTFE	340.92	581.18	Mn=316

We added the following modifications to the main text and method:

Method:

“Fluorinated ethylene propylene (FEP 100, Dupont, Mn =676 000 000), Polytetrafluoroethylene (PTFE 7AX, Dupont, Mn= 316 000 000), Polypropylene (PP, Macklin, Mw = 289 300, (see Supplementary Fig. 22)), High-density polyethylene (HDPE, Sigma-Aldrich, Mw = 3 000 0000 ~ to 6 000 000), Silicon dioxide (SiO₂, Aladdin, 99.95%), p-Benzoquinone (C₆H₄O₂, Macklin, ≥99.5%), Dimethyl sulfoxide (C₂H₆SO, Macklin, 99.8%), Ethylenediaminetetraacetic acid disodium salt (C₁₀H₁₄N₂ Na₂O₈·2H₂O, Aladdin, 99%), Tert-butanol (C₄H₁₀O, Macklin, 98.0%), Silver nitrate (AgNO₃, Macklin, 99.8%), Rhodium(III) nitrate dihydrate (H₄N₃O₁₁Rh, Macklin, 95%), Iridium chloride(III) hydrate (IrCl₃·3H₂O, Macklin, Ir>52%), Mercury nitrate monohydrate (HgN₂O₆·H₂O, Macklin, 99%), Silver acetate (C₂H₃O₂Ag, Macklin, 99.5%), Palladium nitrate dihydrate

($\text{Pd}(\text{NO}_3)_2 \cdot 2\text{H}_2\text{O}$, Macklin, $\text{Pd} \geq 39\%$), Potassium gold chloride (KAuCl_4 , Macklin, $\text{Au} \geq 51\%$), Potassium(II) tetrachloroplatinate (K_2PtCl_4 , Macklin, 98%), 5,5-dimethyl-1-pyrroline N-oxide [$\text{C}_2\text{H}_6\text{OS}$, Dojindo], 2,2,6,6-tetramethylpiperidinoxy (TEMPO, $\text{C}_9\text{H}_{18}\text{NO}$, Macklin, 98%), Sodium chloride (NaCl , Macklin, 99.5%), Hydrochloric acid (HCl , Macklin, AR), Nitric acid (HNO_3 , Macklin, AR), Sodium hydroxide (NaOH , Macklin, 99.5%), DMSO (Macklin, 99.7% Water ≤ 50 ppm), Ethanol (Macklin, 99.7%), Acetonitrile (Sinopharm Chemical Reagent Co. Ltd., 99%). CPUs, plating wastes, and dismantled factory waste machines were purchased from Alibaba”.

“The FEP, PTFE, PP and HDPE were characterized by Fourier Transformed Infra-Red Spectroscopy (FTIR), Thermogravimetric Analysis-Differential Scanning Calorimetry (TG-DSC), and Gel Permeation Chromatography (GPC) (See, Sample Characterization and Supplementary Fig. 22-24)”

[R4] Sperati, C.A. & Starkweather, H.W. Fluorine-containing polymers. II. Polytetrafluoroethylene. In: *FORTSCHRITTE DER HOCHPOLYMEREN-FORSCHUNG*. Advances in Polymer Science, 2/4. Springer, Berlin, Heidelberg (1961).

5. Why Pd was chosen as the model ion for calculation rather Au?

Thank you for your question. When we were writing the manuscript, we first attempted calculations using AuCl_4 and PdCl_4 , with 12 molecules of H_2O ; but all these calculations failed to converge. We then resorted to calculate for Pd alone as results for any metal would be the similar. Below, you can see the results of this calculations for Au. We find that, just like for the case of palladium, the presence of the metal ion tends to bring the oxygen close to the polymers; the bond length decreases from 4.5851 Å to 4.402 Å and finally to 4.256 Å, which is mirrored in the adsorption energy that decreases from -0.0483 eV to -1.2374 eV and reaches down to -1.2714 eV.

Figure R5. Calculated distance between O of oxygen and F, and adsorption energy for PTFE and O₂. Carbon: Purple, Grey: Fluoride, Red: Oxygen, Yellow: Gold.

We added the results of the calculation for Au in the supplementary information and the following comments to the manuscript

Main Text:

*“Similar calculations have been performed for the case of gold; they show similar trends in both the variations of bond length and adsorption energy (see **Supplementary Fig. 18**).”*

6. It's not easy for maintaining N₂ atmosphere constantly in beaker under ultrasound. More details are required for the reaction setup.

Thank you for bringing to our attention that the description of the setup needs to be completed. The setup we present in figure 1 is representing standard beakers but serves only the purpose of illustration. In anaerobic conditions (N₂), we employed a reactor with top mounted outlet and inlet valves; after bubbling the valves are closed before starting the ultrasonication. The reactor is a gas-washing bottle. The seal consists of a PTFE gasket, rubber ring and silica gel grommet.

Figure R6. Schematic of reactor. The seal consists of a PTFE gasket, rubber ring and silica gel grommet.

To take into account your remark we made the following modification to the manuscript in the method section:

Method:

*The catalyst (10 mg) is introduced into a **gas washing** bottle (see **Supplementary Fig. 25**) containing a solution (50 ml) at a certain concentration of metal ions (0.001 mM, 0.01 mM, 0.1 mM, or 1 mM). The reaction system is then ultrasonicated (120 W, 40 kHz) for a given amount of time (1 h, 3 h, or 20 h). The temperature of ultrasonic water was controlled by a thermostat connected to a radiative copper cooler placed in the ultrasonic bath at 25 °C, unless otherwise specified.*

In aerobic conditions, the protocol is followed without needing any further preparation.

Anaerobic conditions are created by saturating the solution with gaseous N₂ gas (15 min). The inlet and outlet valves of the sealed gas washing bottle are then closed. The reactor is finally placed into an ultrasonic bath (40 kHz, 120 W). The solution is saturated again with N₂ after each aliquot (1 mL) has been sampled.”

Reviewer #2 (Remarks to the Author):

All the comments have been well addressed to improve the manuscript. The revised version was suggested to be accepted.

We thank reviewer #2 for their contribution to the previous round of review and that the manuscript was evaluated as ready for acceptance.

REVIEWERS' COMMENTS

Reviewer #1 (Remarks to the Author):

All the comments have been well addressed. This manuscript was suggested to be accepted in the current version .